# DATA-CENTRIC LESSONS TO IMPROVE SPEECH-LANGUAGE PRETRAINING

**Vishaal Udandarao**[1,2,3], **Zhiyun Lu**[1], **Xuankai Chang**[1], **Yongqiang Wang**[1]
**Albin Madapally Jose**[1], **Fartash Faghri**[1], **Josh Gardner, Chung-Cheng Chiu**[1]
[1]Apple    [2]University of Cambridge    [3]University of Tübingen

## ABSTRACT

Spoken Question-Answering (SQA) is a core capability for useful and interactive artificial intelligence systems. Recently, several speech-language models (SpeechLMs) have been released with a specific focus on improving their SQA performance. However, a lack of controlled ablations of pretraining data processing and curation makes it challenging to understand what factors account for performance, despite substantial gains from similar studies in other data modalities. In this work, we address this gap by conducting a data-centric exploration for pretraining SpeechLMs. We focus on three questions fundamental to speech-language pretraining data: (1) how to *process* raw web-crawled audio content for speech-text pretraining, (2) how to *construct* synthetic datasets to augment web-crawled data and (3) how to *interleave* (text, audio) segments into training sequences. We apply the insights from our controlled data-centric ablations to pretrain a 3.8B-parameter SpeechLM, called **SpeLangy**, that outperforms models that are up to 3x larger by 10.2% absolute performance. We hope our findings highlight the impact of effective data curation and guide future data-centric exploration in SpeechLMs.

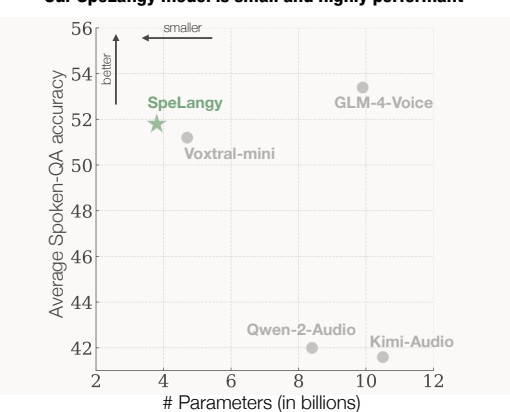

Figure 1: **(Left)** We highlight the *data-centric* questions we study in this work (Sec. 3), **(Right)** Distilling all our data-insights yields a strong 3.8B-parameter SpeechLM, **SpeLangy** (Sec. 5).

## 1 INTRODUCTION

Language-based assistants are now widely deployed (OpenAI, 2024; Comanici et al., 2025). Yet, purely textual interactions are inherently limiting for real-world assistants that must operate in open, hands-free settings. Voice provides a natural, low-friction interface for human–AI interaction, and recent work therefore emphasizes Spoken Question-Answering (SQA) (Nachmani et al., 2023; Liu et al., 2025; Xiaomi, 2025)—where a question is asked in audio and the system must produce spoken or textual answers—as a core capability for end-to-end speech language models (SpeechLMs).

Recently, *speech–text interleaved pretraining*—next-token prediction over sequences that alternate between speech and text tokens—has been proposed as a viable strategy to boost SQA performance (Nguyen et al., 2025b; Zeng et al., 2024b). However, while these works describe modeling choices

comprehensively, details of their data pipelines are often not evaluated in a controlled setting. How should we process raw audio into trainable speech-text chunks? Can we leverage text-only datasets to go beyond datasets sourced from raw audio? How should we interleave tokens for effective modality alignment? In the current literature, *these data-centric questions remain underexplored*. In other domains like language (Dubey et al., 2024; Li et al., 2024) and vision (Gadre et al., 2023; Siméoni et al., 2025), data curation has consistently proven to be a primary driver of performance improvements, yet a large gap exists from the data-centric perspective in the speech-language domain.

In our work, we aim to close this gap with a systematic, *data-centric* study of interleaved pretraining for SQA (Fig. 1). To operationalize our goal, we use a controlled study design that only uses a speech-text interleaving task during pretraining, thereby removing the confounders of task interference, suboptimal data-mixing ratios etc. that other SpeechLM pretraining pipelines (Ding et al., 2025; Zeng et al., 2024a; Li et al., 2025c; Xiaomi, 2025) often suffer from. Our experimental methodology is inspired by recent data-centric works that emphasize the importance of clean empirical setups for conducting controlled data ablation experiments focused on a single modality or setting (Li et al., 2024; Gadre et al., 2023). To the best of our knowledge, ours is the first work to systematically compare different data strategies for speech-language interleaving strategies, on a level-playing field.

We first provide a detailed description of our processing pipeline for converting raw audio into speech-text interleaved data (Fig. 9). We then study optimal interleaving strategies for speech-text pretraining, finding that fine-grained interleaving (which alternates between speech and text modalities at sentence boundaries) improves alignment of the two modalities (Sec. 3.3). Building on this, we introduce effective synthetic data methods involving LLM-based rewriting and text-to-speech synthesis to go beyond raw web-crawled audio for pretraining (Sec. 3.4). We also examine two modality-sampling schemes for interleaved training, finding that a deterministic ordering of alternating speech-text chunks is beneficial compared to stochastic modality sampling (Sec. 3.5). Further, we show our pretraining data interventions also improve models under the audio-understanding only setting (Sec. 3.6) and after post-training (Sec. 3.7). To understand *why* our data-centric methods improve performance, we analyse the modality gap between speech and text distributions (Sec. 4.1) and inspect the topic distributions of web-crawled and synthetic datasets (Sec. 4.2). Finally, to showcase the efficacy of our data interventions at scale, we pretrain a 3.8B SpeechLM (**SpeLangy**) that outperforms 3x larger models by 10% average SQA performance, across three standard benchmarks. Taken together, our results underscore the central role of data curation in speech–language pretraining and motivate a broader, systematic push toward data-centric exploration.

## 2 RELATED WORK

**Speech Language Models.** Most SpeechLMs employ a simple Speech Encoder + Connector + LLM philosophy for joint speech-text training (Lakhotia et al., 2021; Algayres et al., 2023; Hassid et al., 2023; Nguyen et al., 2025b; Nachmani et al., 2023; Rubenstein et al., 2023; Zhang et al., 2023; Défossez et al., 2024; Liu et al., 2025). Models like Kimi-Audio (Ding et al., 2025), Step-Audio-2 (Wu et al., 2025a), Baichuan-Audio (Li et al., 2025c), GLM-4-Voice (Zeng et al., 2024a), and MiMo-Audio (Xiaomi, 2025) have emerged as strong models that seamlessly perform several tasks, including spoken question-answering. While demonstrating impressive performance, details behind their data curation strategies are however scant. Through our controlled experiments, we fill this gap in the SpeechLM domain by showcasing how to effectively construct speech-text pretraining datasets.

**Data Curation for Foundation Models.** Pretraining data quality is pivotal for driving performance of foundation models. Efforts like Gopher (Rae et al., 2021), T5 (Raffel et al., 2020), Nemotron-CC (Su et al., 2024), FineWeb (Penedo et al., 2024), DCLM (Li et al., 2024) and OLMo-2 (OLMo et al., 2024) significantly emphasize the benefits of strong data processing, curation and filtering for language data. In computer vision, Dinov2 (Oquab et al., 2023), Dinov3 (Siméoni et al., 2025), AIMv2 (Fini et al., 2025) and Web-SSL (Fan et al., 2025) showcased the high impact that careful data curation has on model quality. Similar results on the importance of data-centric research have been shown in vision-language (Gadre et al., 2023; Fang et al., 2023a; Tong et al., 2024a; Wang et al., 2025b) and reasoning-based (Guha et al., 2025; Li et al., 2025d; Muennighoff et al., 2025) foundation modeling literature. Owing to the paucity of such data-centric research in the speech-language domain, we aim to close this gap through a set of controlled data ablations, demonstrating the strong utility of data-centric approaches for boosting SpeechLM quality.

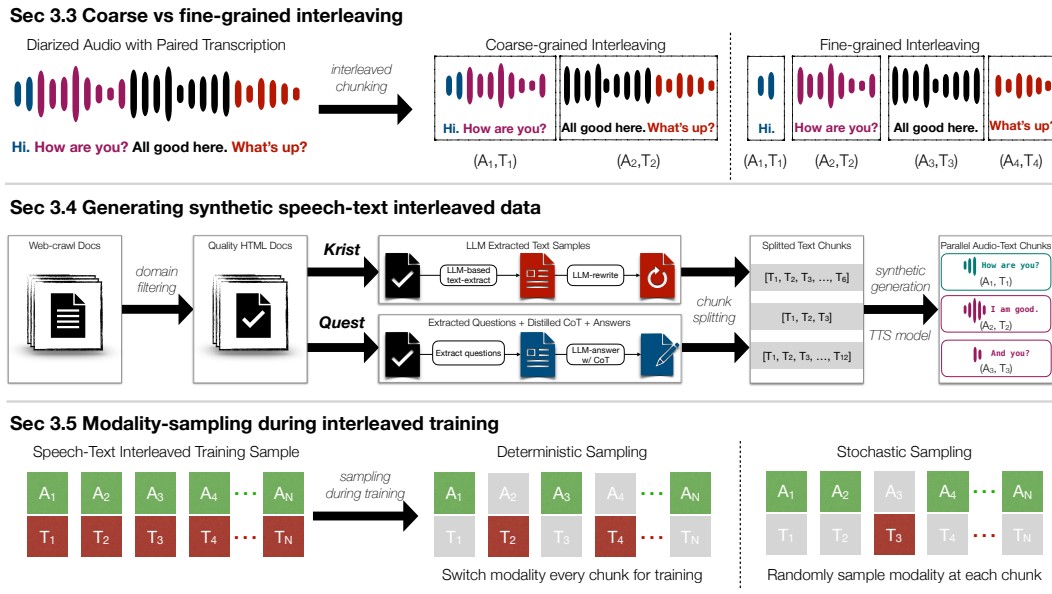

Figure 2: **Our experimental conditions for speech-text pretraining data.** (**Top**) We study two interleaving strategies: *coarse* (long chunks) and *fine* (short chunks) (Sec. 3.3). (**Middle**) We construct two synthetic datasets—*Krist* and *Quest*—from filtered knowledge-rich web-documents (Sec. 3.4). (**Bottom**) We study two schemes for interleaved training: *deterministic* and *stochastic* (Sec. 3.5).

# 3 CONTROLLED DATA-CENTRIC EXPERIMENTS

In this section, we address our three key *data-centric* questions for improving SQA, via controlled experiments: (1) how to *process* raw web-crawled audio into suitable interleaved speech-text training data (Sec. 3.3), (2) how to *construct* synthetic speech-text datasets seeded from text-only datasets (Sec. 3.4), and (3) how to *interleave* between speech and text modalities while training (Sec. 3.5).

## 3.1 EVALUATION BENCHMARKS

**Spoken Question-Answering (S→T).** We use three standard benchmarks for SQA where the model is asked questions in speech and is tasked to respond in text (S→T): *Spoken-LLaMA-Questions* (SLQ), *Spoken-Web-Questions* (SWQ) and *Spoken-TriviaQA* (STQ). We source all the audio questions from OpenAudioBench (Li et al., 2025c). Our protocol follows standard language modeling pretraining evaluations (Gu et al., 2024; Allal et al., 2025) to use an MCQ cloze-format with log-likelihood evaluation for choosing the correct option (we use 4 multiple choices with chance-level accuracy being 25%). We provide more details and examples from each of our evaluation datasets in Appx. G.

**Text Understanding (T→T).** To ensure our pretraining recipe does not degrade base LM performance, we evaluate on 12 standard benchmarks spanning general knowledge, math and coding: *MMLU* (Hendrycks et al., 2020), *CoreEN* (Gunter et al., 2024; Mizrahi et al., 2025; Busbridge et al., 2025) (consisting of 9 benchmarks—*ARC-Easy* and *ARC-Challenge* (Clark et al., 2018), *HellaSwag* (Zellers et al., 2019), *Lambada* (Paperno et al., 2016), *PIQA* (Bisk et al., 2020), *SciQ* (Welbl et al., 2017), *TriviaQA* (Joshi et al., 2017), *WebQuestions* (Berant et al., 2013), and *WinoGrande* (Sakaguchi et al., 2021)), *GSM-8k* (Cobbe et al., 2021), and *HumanEval* (Chen et al., 2021).

## 3.2 BASE SETUP

**Model Architecture.** We conduct all our experiments with a ∼3.8B-parameter SpeechLM, consisting of two major components: a speech tokenizer and a pretrained language model. Our speech tokenizer consists of a 1B-param speech encoder with conformer (Gulati et al., 2020) blocks with 8x downsampling followed by a finite scalar quantizer (Mentzer et al., 2023) that outputs discrete speech tokens at 80ms per token (12.5Hz). The speech tokenizer is trained jointly with a combination of ASR and reconstruction loss, to jointly optimize phonetic and higher-level structure. We initialize our

language model with the dense 2.8B base-LM from (Li et al., 2025b) that has a context-length of $16,384$ tokens. The LM we start from has undergone no additional continued-pretraining. The LM does not support speech tokens natively. We extend the vocabulary to include speech tokens. We initialize the new token embeddings with Xavier normal initialization (Glorot & Bengio, 2010).

**Training Data.** Our base data mixture consists of web-crawled audio that we process into interleaved speech-text data. We provide more details on how we process audio into our training data format in the next section. We also use the text continued-pretraining dataset from (Li et al., 2025b) to preserve the base-LM's text performance. Following prior works (Shukor et al., 2025; McKinzie et al., 2024), we use a $60\%$ text-only and $40\%$ speech-text data mixture during interleaved pretraining.

**Optimization Details.** We train with global-batch-size of $512$ and packed-sequence-length of $16,384$ tokens, for 200k steps. We use standard next-token prediction objective and compute loss over both speech and text tokens (we also ablate with loss-masking on speech tokens in Sec. 3.6). We only tune language model while keeping speech tokenizer frozen. For more details, refer to Appx. E.

### 3.3 PROCESSING PRETRAINING DATA VIA FINE-GRAINED INTERLEAVING

**Extracting interleaved data from raw audio.** We begin with >10M hours of raw web-crawled audio. To process them into trainable speech-text samples, we follow a multi-stage pipeline (see Fig. 9 in Appendix), involving *speaker diarization*, *language detection and filtering*, *paired-transcription generation and filtering*, and *interleaved chunking*. Our pipeline yields interleaved training samples $X_i$ consisting of multiple paired speech-text chunks of the form $X_i = \{(A_1,T_1),(A_2,T_2)\cdots(A_n,T_n)\}$, where $n$ is the number of chunks in each sample. We provide more details about each individual component along with stats in Appx. A, while focusing on the *interleaved chunking* component here.

**Fine vs coarse interleaving.** Prior speech-text pretraining works (Liu et al., 2025; Zeng et al., 2024a) have explored constructing interleaved data from raw audio. However, they do not quantify the importance of *interleaving granularity* for effective training. To study this, we construct two interleaving variants (see Fig. 2-A)—(1) *coarse interleaving*, where we merge multiple consecutive diarized outputs into one if tagged with same speaker-ID, yielding long chunks, and (2) *fine interleaving*, where we keep all diarized outputs as is without merging, yielding short chunks. As expected, from Fig. 3, we find coarse interleaving leads to longer chunks (mean-length=19.2s) compared to fine interleaving (mean-length=5.2s). From Tab. 1, we note fine interleaving improves SQA performance by $3.1\%$ on average, while matching text-only performance. This is a significant finding since the default approach in prior works (Ding et al., 2025; Li et al., 2025c) has been to merge

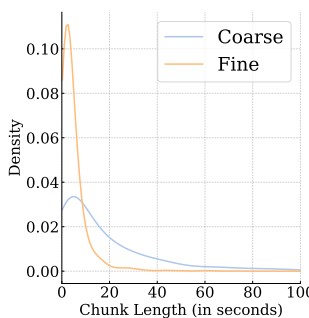

Figure 3: Audio chunk length distribution (in seconds) for our interleaving strategies.

same-speaker diarization outputs, yet our results advocate for more granular interleaving. Hence, for all our experiments, we adopt fine interleaving for web-crawled speech-text pretraining by default.

Table 1: **Fine interleaving improves over coarse interleaving.**

| Interleaving Granularity | Text Understanding (T→T) | | SQA (S→T) acc (%) | | | |
|---|---|---|---|---|---|---|
| | *CoreEN* | *MMLU* | *SWQ* | *STQ* | *SLQ* | *Avg* |
| *Text-init* (no-speech) | 62.4 | 62.2 | – | – | – | – |
| Coarse | **60.4** | 63.9 | 42.5 | 26.6 | 43.6 | 37.6 |
| Fine | **60.4** | **64.1** | **42.7** | **32.2** | **47.3** | **40.7** |

**Takeaway:** Fine-grained interleaving of speech-text pretraining data boosts SQA performance.

### 3.4 CONSTRUCTING EFFECTIVE SYNTHETIC DATASETS

While web-crawled datasets offer massive volume, they often have poor *domain coverage*—their data distribution does not reflect the highest-priority domains for downstream deployment (Baack, 2024; Longpre et al., 2024). Often, sufficient data from many core domains simply does not exist or is hard to crawl (Zhang et al., 2024c; Fang et al., 2023b; Kydlíček et al., 2025). Together, these motivate

using synthetic data to augment existing web-crawl data. Moreover, in our web-crawled audio data, we find noisy text-annotations (due to hallucinations from transcription models) and artifacts like background noise and speaker overlap. Thereby, we explore synthesizing clean speech-text datasets from existing text-only corpora. We build two synthetic datasets (see Fig. 2-B)—Knowledge-Rich Interleaved Speech-Text (*Krist*) and Question-Answering Speech-Text (*Quest*).

**Knowledge-Rich Interleaved Speech-Text (Krist).** We start from lightly-filtered web-crawled documents (similar to WARC files from CommonCrawl (2007)). We then apply URL-filtering to preserve documents from *knowledge-rich domains* (list of domains is in Appx. C.1). This is motivated by recent efforts advocating high-quality educational data for accelerating model training (Penedo et al., 2024; Abdin et al., 2024; Gunasekar et al., 2023). Next, we use gpt-4o-mini to extract and lightly rewrite the text-content from raw HTML, following Maini et al. (2024) (prompt used in Appx. C.2). We then segment the texts based on sentence-level splitting, to produce different text chunks. Finally, we synthesize audio for each chunk using melo-TTS (Zhao et al., 2023). To improve speaker diversity in the synthesized data, we randomly sample voices from 5 different accents. This pipeline yields ∼4.6M hours of interleaved speech-text data.

**Question-Answering Speech-Text (Quest).** Since *Krist* is synthesized from HTML-extracted text, its samples do not sound natural. We therefore build *Quest*, explicitly organized in question-answering format to mimic real audio. Starting from the same high-quality HTML pool as *Krist*, we first mine all possible question texts using regex-parsing. We then use gpt-4o to filter out invalid questions (some examples in Appx. C.3). Finally, we use gpt-4o to generate responses along with a chain-of-thought (Wei et al., 2022) trace (generation prompts in Appx. C.2). We use the same sentence-level chunking strategy as *Krist*. This pipeline produces ∼0.9M hours of interleaved speech-text data.

Table 2: **Synthetic speech-text interleaved data improves over web-crawl data.**

| Data Mix | Text Understanding (T→T) | | SQA (S→T) acc (%) | | | |
|---|---|---|---|---|---|---|
| | *CoreEN* | *MMLU* | *SWQ* | *STQ* | *SLQ* | *Avg* |
| *Text-init* (no-speech) | 62.4 | 62.2 | – | – | – | – |
| Web-crawl 100% | 60.4 | 64.1 | 42.7 | 32.2 | 47.3 | 40.7 |
| Web-crawl 53% + Krist 47% | **60.8** | 64.8 | 43.4 | 29.2 | 52.0 | 41.5 |
| Web-crawl 66% + Quest 34% | 60.4 | **66.2** | 42.7 | **34.7** | **66.3** | **47.9** |
| Web-crawl 59% + Quest 6% + Krist 35% | 60.7 | 65.9 | **43.8** | 31.5 | 51.0 | 42.1 |
| Web-crawl 40% + Quest 27% + Krist 33% | 60.6 | 65.7 | 43.3 | 31.7 | 49.3 | 41.4 |

**Results.** We study the impact of independently mixing *Krist* and *Quest* with web-crawled data (mixed proportional to their approximate token counts, for details see Appx. D) in Tab. 2. We find mixing in *Krist* brings a 0.8% lift in SQA performance while also moderately benefitting text-only benchmarks, compared to training on web-crawl alone. Further, mixing *Quest* with web-crawl improves both MMLU and SQA performance by large margins of 2.1% and 7.2%. We hypothesize that the QA format in interleaved training with *Quest* helps to efficiently adapt to downstream SQA capabilities. We additionally explore two ratios for mixing *Quest* and *Krist* with the web-crawled data—one where we sample according to approximate token-counts of each data source (59% web-crawl), and another where we upsample the synthetic proportion (40% web-crawl). Both settings improve over web-crawl by 1.4−0.7% SQA. However, due to complex interactions between mixing ratios and data repeats (Muennighoff et al., 2023; Xue et al., 2023), it is unclear how to construct an optimal mixture extracting the best of each data source (Shukor et al., 2025; Ye et al., 2024a) (details on exact token counts in Appx. D). We leave such a data-mixing exploration for future work.

> **Takeaway:** Synthetic datasets using TTS models bring gains when mixed with web-crawled data.

### 3.5 MODALITY SAMPLING SCHEMES FOR INTERLEAVED TRAINING

So far, we have discussed interleaved speech-text data *processing* and *curation* for improving SQA performance. However, we did not describe *how we sample modality chunks during interleaved training*. Here, we study two sampling schemes as shown in Fig. 2-C. Recollect that each interleaved training sample is of form $X_i = \{(A_1, T_1), (A_2, T_2) \cdots (A_n, T_n)\}$. We now test two variants:

**Stochastic Sampling.** In the first variant (used in all our previous experiments), at each chunk $i$, we randomly sample the chunk-modality with $0.5$ probability. The modality sampling at each chunk $i$ is independent of all other chunks $j{\neq}i$. We always start with an audio chunk $A_1$, to ensure that there is at least 1 audio chunk in our training sequence.

**Deterministic Sampling.** While the stochastic variant allows flexibility and potentially offers better generalization, it can restrict the number of *modality switches* during training. Hence, we test a deterministic approach, where we alternate between audio and text modalities at each chunk, i.e. we formulate the training sequence as $\{A_1, T_2, A_3 \cdots A_{n-1}, T_n\}$. This *maximizes the number of modality switches* for a given sample. Here too, we always start with $A_1$.

**Results.** From Tab. 3, we find deterministic sampling boosts SQA performance by 1% on average over stochastic sampling. We posit that the number of modality switches during training affects the SQA performance—in Fig. 4, we plot the distribution of modality switches occuring during interleaved training, finding that stochastic sampling switches modalities quite infrequently, whereas the deterministic approach has a higher number of modality switches during training. Indeed, the expected number of modality switches for a sample consisting of $n$ chunks is $n{-}1$ for deterministic sampling and $\frac{n-1}{2}$ for stochastic sampling. By frequently switching modalities more often, deterministic sampling likely enables more effective cross-modal learning, thereby improving downstream SQA performance.

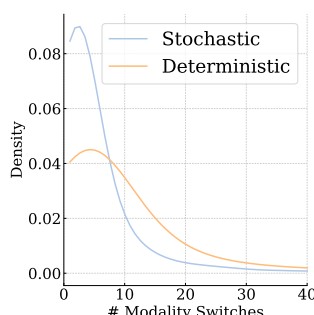

Figure 4: Modality switches during interleaved training for our two sampling schemes.

Table 3: **Deterministic speech-text sampling improves over stochastic sampling.**

| Sampling scheme | Text Understanding (T→T) | | SQA (S→T) acc (%) | | | |
|---|---|---|---|---|---|---|
| | *CoreEN* | *MMLU* | *SWQ* | *STQ* | *SLQ* | *Avg* |
| *Text-init* (no-speech) | 62.4 | 62.2 | – | – | – | – |
| Stochastic | **60.6** | **65.7** | 43.3 | **31.7** | 49.3 | 41.4 |
| Deterministic | 60.1 | 65.2 | **44.2** | 31.2 | **51.7** | **42.4** |

> **Takeaway:** Deterministic sampling improves SQA over stochastic for interleaved training.

## 3.6 OUR DATA-CENTRIC LESSONS TRANSFER TO UNDERSTANDING-ONLY SPEECHLMS

So far, we showed our data-centric methods boost SQA significantly. These results were achieved while computing loss on both audio and text tokens during interleaved training to support a native end-to-end SpeechLM. However, there is also great interest in developing an understanding-only SpeechLM that ingests both audio and text and outputs only text, e.g. the Thinker model in the Thinker-Talker architecture series (Xu et al., 2025). In this vein, many prior works (Liu et al., 2025; Li et al., 2025c) apply loss masking on the audio tokens while doing speech-text interleaved training.

Table 4: **Our data-centric methods also work for understanding-only SpeechLM**

| Method | Text Understanding (T→T) | | SQA (S→T) acc (%) | | | |
|---|---|---|---|---|---|---|
| | *CoreEN* | *MMLU* | *SWQ* | *STQ* | *SLQ* | *Avg* |
| Baseline (w/o loss-masking) | 60.4 | 63.9 | 42.5 | 26.6 | 43.6 | 40.7 |
| + all data interventions | 60.1 | 65.2 | 44.2 | 31.2 | 51.7 | 42.4 |
| Baseline (w/ loss-masking) | 61.7 | 66.5 | 45.9 | 34.0 | 47.7 | 42.5 |
| + all data interventions | **61.8** | **67.3** | 45.7 | **44.6** | **65.0** | **51.8** |

We hence test if our three data strategies also transfer to this audio-loss-masked setting. From Tab. 4, we find this indeed to be the case (9.3% average SQA lift). Further, we find absolute SQA performance improves significantly with loss-masking (51.8% with loss masking vs. 42.4% without). This result

corroborates prior results (Liu et al., 2025; Li et al., 2025c; Chu et al., 2024) suggesting that, for small scale models there is an inherent modality conflict between audio and text tokens, which can lead to regressions when computing loss on both speech and text modalities.

> **Takeaway:** Our three data interventions also transfer to the understanding-only SpeechLM setting.

### 3.7 OUR DATA-CENTRIC LESSONS TRANSFER AFTER POST-TRAINING

Previously, our methods were only tested for speech-text interleaved pretraining. Our models are hence inherently base models, and cannot be used in an assistant-like manner. However, due to the importance of real-world-assistant use-cases, we test if our gains also hold after instruction-tuning.

**Post-training setup.** We started from our base model checkpoints and conducted supervised fine-tuning (SFT), with a data-mix of QA conversations, TTS and ASR-style conversations. For more details on SFT data, refer to Appx. M. We selected 3 checkpoints from our previous experiments to conduct SFT training using *exact same SFT data*. The first, denoted as coarse, is trained on web-crawl only data with coarse interleaving (Row 1 in Tab. 1); second, denoted as fine, is trained on web-crawl data with fine interleaving (Row 2 in Tab. 1 and Row 1 in Tab. 2); third is the best model in Tab. 2, denoted as fine+syn, trained on web-crawl and *Quest* (Row 3 in Tab. 2).

**Evaluations.** We evaluate *text response quality and audio response quality*. To evaluate text response quality, we use *spoken-alpaca* and *noisy-alpaca*. For audio response quality, we use 5 datasets from third-party vendors and use LLM-as-judge. For more details on eval setup, refer Appx. M.3.

**Results.** From Tab. 5, we observe that the gains obtained from our pretraining data interventions are largely carried on to the SFT stage, for both text and audio metrics. This suggests that SQA accuracy can be a good proxy metric for model quality after post-training as well. Similar results suggesting that front-loading high quality data into pretraining can benefit post-trained models have been shown in the text-only domain (Akter et al., 2025; Shah et al., 2025). Taken together, our results demonstrate the effectiveness of our proposed data-centric methods on downstream SFT tasks.

Table 5: **Comparison of model's text/audio response quality after SFT.**

| Pretrain ckpt | Text Quality[1] | | Audio Quality[2] | | | | |
|---|---|---|---|---|---|---|---|
| | spoken-alpaca | noisy-alpaca | Eval 1 | Eval 2 | Eval 3 | Eval 4 | Eval 5 |
| coarse | 42.6 | 45.2 | 37.4 (17.2) | 33.3 (24.1) | 34.3 (18.1) | 37.0 (16.3) | 38.8 (16.9) |
| fine | 44.3 | 47.3 | 39.9 (18.5) | 33.8 (23.7) | 36.4 (11.6) | 38.0 (16.9) | **41.9 (20.7)** |
| fine + syn | **47.4** | **48.8** | **41.1 (17.1)** | **36.6** (23.1) | **40.1** (18.7) | **39.4** (16.9) | 39.3 (16.8) |

> **Takeaway:** Our three data-centric pretraining methods also improve post-trained SFT checkpoints.

## 4 UNDERSTANDING WHY OUR DATA INTERVENTIONS HELP

### 4.1 IMPROVED ALIGNMENT BETWEEN MODALITY DISTRIBUTIONS

Here, we aim to better understand why our data interventions (fine chunking + synthetic data mixing) improve over a baseline with coarse chunking and no synthetic data. One plausible hypothesis is that *fine interleaving and synthetic data close the gap between the model's audio-conditioned output distribution and text-conditioned output distribution*. Since we initialize from a well-trained language model, ensuring the audio-conditioned output distribution matches the distribution of text-conditioned outputs enables *strong modality alignment*. We now test if our approaches close this gap.

**Setup.** We start with the Spoken-LLaMA-Questions test set. For each sample, we independently compute the token-wise teacher-forced probability distributions based on conditioning on audio and text questions separately. We then compute the mean token-wise reverse-KL-divergence values between the probability distributions. For details, please refer to Appx. J.

---

[1]Length-controlled (Dubois et al., 2024) win rates in % against the reference model.
[2]Win (Tie) rates in % against the reference model.

**Results.** In Fig. 5, we plot the distribution of mean reverse-KL-divergence values between text-conditioned and audio-conditioned output distributions on the full Spoken-LLaMA-Questions test set (see Appx. J for definition of reverse-KLD). We find that fine interleaving induces lower KL-divergence values (mean=2.21) compared to coarse interleaving (mean=3.20). Moreover, a model trained with both fine interleaving and synthetic data further closes the modality distribution gap (mean KLD=1.47). This trend also holds across other metrics (see Appx. J). This suggests that our data interventions indeed close the gap between text-conditioned and audio-conditioned probability distributions, thereby better aligning the two modalities, leading to stronger downstream SQA performance.

Figure 5: Our methods reduce distribution gap (reverse-KLD) between text and audio.

## 4.2 Synthetic data improves domain coverage

Previously in Sec. 3.4, we observed that our synthetic speech-text datasets improve both text and SQA performance significantly. Our central hypothesis for *why* is—*web-crawled data has a very skewed topic distribution and our synthetic data improves the domain coverage*. To help understand the composition of our web-crawled and synthetic datasets from a *topic* perspective, we leveraged the topic-domain classifier from (Wettig et al., 2025)[3], which can categorize texts in 24 different topic domains (an analysis with more fine-grained classifiers is in Appx. K.3). We run the classifier on 5000 random samples from each of our training datasets (*Web-crawl*, *Krist* and *Quest*). We also annotate topics in evaluation datasets. From Fig. 6 (more results in Appx. K), we make two observations:

- **Web-crawled data is highly skewed** and is majorly comprised of *entertainment*, *sports and fitness*, *religion* and *social life* domains. This is not surprising given that most of our web-crawled audio data is sourced from podcasts, interviews, talk-shows and monologues.

- **Synthetic data improves topic coverage.** It is evident that both the *Krist* and *Quest* datasets oversample data from the domains of *science and tech*, *health*, *education and jobs*, and *finance*, all of which are extremely under-represented in the web-crawled data.

Therefore, by enabling broader coverage of topic domains, our synthetic datasets help to (1) close the distribution mismatch between the raw web-crawled data and the downstream evaluation datasets, and (2) enhance the diversity of our pretraining data distribution. Our findings extend prior work in the language space that have discussed the importance of training data diversity and domain coverage (Nguyen et al., 2025a; Maini et al., 2025; 2024) to the speech-language domain.

## 4.3 Analysing train-test contamination

Given the significant boosts induced by our synthetic datasets, a natural question arises—*Is there test-set leakage, and if so, how does it impact SQA performance?* To address this, we conduct a contamination analysis with two goals in mind: (1) identify proportion of test samples that are likely contaminated in our training data, and (2) understand the performance impact of this leakage.

**Contamination detection.** To find the extent of contamination in our synthetic datasets, we follow recent works (Singh et al., 2024; Sainz et al., 2024; Dubey et al., 2024) and use $n$-gram token overlaps. While prior works used $n$=13, we opt for a window from $n$=6 to $n$=13 to improve recall, at the expense of more false-positives. We use the gpt-4o tokenizer and apply lower-case normalization pre-tokenizing. We mark a test sample as contami-

Figure 7: **Proportion of contamination.**

| Eval | % Contamination [# samples] | | |
|------|------|------|------|
| | *Krist* | *Quest* | *All* |
| SWQ | 0.4% [4] | 0.1% [1] | 0.4% [4] |
| STQ | 2.2% [22] | 0.8% [8] | 2.5% [25] |
| SLQ | 6.7% [20] | 0.2% [5] | 7.7% [23] |

nated if we find a matching $n$-gram in any equivalent $n$-token span of a synthetic dataset (pseudo-code in Alg. 1). We consider all three SQA test sets for analysis, and concatenate the question and answer of each sample for matching. For train sets, we take samples from seed text-datasets (from which we synthesize audio) for detecting matches.

---

[3]https://huggingface.co/WebOrganizer/TopicClassifier-NoURL

## Distributions of topic domains in our evaluation and training datasets

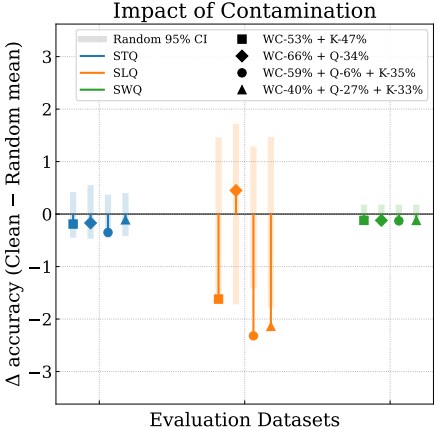

Figure 6: **Synthetic data improves domain coverage.** We plot the distribution of topic domains in our evaluation datasets (in blue, Spoken-LLaMA-Questions (*left*) and Spoken-TriviaQA (*right*)) and contrast them with the topic distribution of web-crawled and synthetic datasets. Our synthetic datasets (*Krist* and *Quest*) fill gaps in domains that are under-represented in the web-crawled data, reducing distribution mismatch, thereby improving SQA performance.

**Proportion of contamination.** We report the proportion of contaminated test samples in Fig. 7. We find that the *Quest* dataset has almost no contamination, while *Krist* has a small, yet non-negligible amount of contamination. Overall, the *SWQ* eval dataset is barely contaminated ($0.4\%$) while *STQ* and *SLQ* evals have $2.5\%$ and $7.7\%$ contaminated samples respectively. Importantly, note that due to our windowed $n$-gram approach, we have many false-positive matches (examples of matches are in Appx. L.1). However, we keep all matches to be as conservative as possible while analysing effect of contamination. To understand the impact of test-set contamination on downstream SQA model performance, we consider the tests sets with these contaminated samples removed as *clean* sets—*SWQ-clean* has 996 samples, *STQ-clean* has 975 samples, and *SLQ-clean* has 277 samples.

**Significance testing setup.** We conduct one-sided significance test on differences b/w performance on *full* test set and *clean* set (removing all contamination). To control for the accuracy difference induced by reducing test set size, we compute *random removal baseline accuracy*—performance after removing same number of randomly selected samples, averaged across 100 bootstrap replicates. We compute empirical $p$-values by comparing clean accuracy against bootstrapped removal distribution. For more details, refer Appx. L.3.

**Results.** We apply significance testing for all 4 models in Sec. 3.4 that use synthetic data. Fig. 8 shows differences b/w *clean* and *random removal mean* for all eval-model pairs, finding contamination does not improve performance for *Spoken-TriviaQA* and *Spoken-Web-Questions*. For *Spoken-LLaMA-Questions*, contamination has minor effect ($1.4-2.1\%$) when *Krist* is used. However, the effect is not statistically significant (significance level of $\alpha=0.01$). We provide more analysis

Figure 8: Differences b/w clean and random removal accuracies with $95\%$ CIs, suggesting contamination has minor effect.

in Appx. L.3. Additionally, we note the performance boosts on SLQ due to synthetic data in Sec. 3.4 ($3.7\%-19\%$) far exceed clean vs random-removal-mean accuracy differences observed (upto $2\%$). Taken together, test-set contamination does not play major role in explaining accuracy boosts.

## 5 SPELANGY: BRINGING IT ALL TOGETHER

Equipped with our key data-centric insights from the previous sections, we now train a 3.8B SpeechLM, called SpeLangy. We use the same training configuration as before, with $16,384$ sequence length trained for 1.67T speech-text tokens. We compare against SoTA speech-language base models including Kimi-Audio (Ding et al., 2025), Qwen-Audio (Chu et al., 2023), and Qwen2-Audio (Chu et al., 2024). We additionally compare two post-trained models—Voxtral-mini (Liu et al., 2025) and GLM-4-Voice (Zeng et al., 2024a)—with the caveat that having undergone instruction-tuning, they are not directly comparable to base models (Dominguez-Olmedo et al., 2024). To ensure our training recipe does not degrade language performance, we also compare against strong open-weights base language models on standard text-only benchmarks.

Table 6: **Spoken Question-Answering (S→T) comparison.** We report results for SoTA SpeechLMs and SpeLangy. Where possible, we report results using pretrained base models (if no base models are released, we evaluate post-trained checkpoints and make a note of this in the table).

| Type | Model | # Params | SWQ | STQ | SLQ | Average |
|------|-------|----------|-----|-----|-----|---------|
| **Base** | Kimi-Audio | 10.5B | 44.0 | 33.8 | 47.0 | 41.6 |
| | Qwen-Audio | 8.4B | **45.7** | 30.3 | 46.0 | 40.7 |
| | Qwen-2-Audio | 8.4B | **45.7** | 33.4 | 47.0 | 42.0 |
| | **SpeLangy** | 3.8B | **45.7** | **44.6** | **65.0** | **51.8** |
| **SFT** | Voxtral-mini | 4.7B | 41.6 | 46.6 | 65.3 | 51.2 |
| | GLM-4-Voice | 9.9B | 43.3 | 52.4 | 64.7 | 53.4 |

Table 7: **Text Understanding (T→T) comparison.** We compare with leading text-only models of same size-class. *Text-init* is the model we start continued-pretraining. Our model is competitive with all compared models, highlighting strong preservation of text-only abilities after speech-text training.

| Model | # Params | CoreEN | MMLU | GSM8k | HumanEval |
|-------|----------|--------|------|-------|-----------|
| *Text-init* | 2.8B | **62.4** | 62.2 | 47.1 | 29.9 |
| Gemma-2 | 2.6B | – | 56.1 | 30.3 | 19.5 |
| Gemma-3 | 4B | – | 62.8 | 38.4 | 36.0 |
| Qwen-2.5 | 3B | – | 65.6 | **79.1** | **42.1** |
| **SpeLangy** | 3.8B | 61.8 | **67.3** | 71.9 | 37.6 |

**Results.** From Tab. 6 we find that our SpeLangy outperforms Kimi-Audio, Qwen-Audio and Qwen-2-Audio by 10.2%, 11.1% and 9.8% on average across the three SQA benchmarks, while being $2.8\times$, $2.2\times$ and $2.2\times$ smaller in size. Further, we obtain competitive performance with the strongly post-trained Voxtral-mini and GLM-4-Voice, *without having undergone any task-specific instruction-tuning*. In Tab. 7, we compare the text performance of SpeLangy with the base LM that we initialize from—we observe large boosts across the board compared to the base-LM, indicating positive text-capability transfer. Further, our model is competitive with Gemma-2 (Team et al., 2024), Gemma-3 (Team et al., 2025) and Qwen-2.5 (Yang et al., 2024) models, all of which are leading open-weights text-only models, highlighting the strength of our SpeLangy model.

## 6 CONCLUSION

In this work, we studied three data-curation methods for speech-language interleaved pretraining to enhance spoken question-answering (SQA) capabilities. We found fine-grained interleaving of speech-text chunks brings large gains, while synthetic datasets synthesized from knowledge-rich seed text-datasets also boosted performance. Deterministic sampling of speech-text chunks during interleaved pretraining further improved SQA results. We showed that these data-centric recipes strengthen alignment between the speech and text modalities and broaden domain coverage of pretraining datasets. Distilling these insights, we pretrained SpeLangy, achieving competitive performance with larger models. We hope our insights motivate more data-centric SpeechLM work.

ETHICS STATEMENT

Our paper leverages web-crawled data for pretraining. Below, we specify how our data collection complies with copyright, licensing, and other web-crawling policies. We further provide details on data provenance, consent, and compliance with legal and ethical standards (e.g., GDPR). Note that the following points are applicable only for our three training datasets: Web-crawl, Krist and Quest. For all other eval datasets, we use publicly open-sourced data and follow their respective usage policies.

1. **Data provenance.** All speech data comes from publicly available podcast RSS feeds and similar spoken-word streams. We do not scrape behind paywalls and avoid clearly copyrighted catalogue content such as commercial audiobooks and music albums.

2. **Web-crawling policies.** Our collection framework respects `robots.txt` directives and website-specific terms of use.

3. **Licensing.** We preferentially include sources under permissive or podcast-typical licenses that allow redistribution for research. When license information is ambiguous, we err on the side of exclusion.

4. **Privacy and PII.** We apply automatic filters to reduce personally identifiable information (e.g., email addresses, phone numbers) and run safety classifiers to remove clearly harmful or sensitive content. We have used this data under our institution's data processing policies.

5. **GDPR and regional compliance.** Processing is conducted in accordance with internal legal guidance, and only aggregate, non-identifiable statistics are reported. No attempt is made to profile or target individual speakers.

ACKNOWLEDGEMENTS

The authors would like to thank (in no particular order): Xiang Kong, Haoxuan You, Zijin Gu, Dongseong Hwang, Tom Gunter, Floris Weers, David Mizrahi, Surabhi S Nath and Thao Nguyen for helpful feedback at various stages of the project. This was work done while VU was at Apple.

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

APPENDIX

## A  PREPROCESSING WEB-CRAWLED AUDIO AS INTERLEAVED TRAINING DATA

In this section, we provide more details about each step in our data processing pipeline for converting web-crawled audio into interleaved speech-text format. We highlight all the components in Fig. 9.

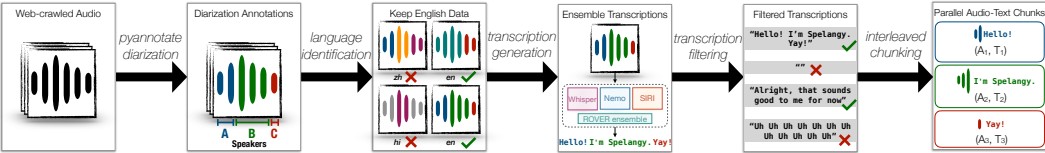

Figure 9: Our processing pipeline to convert raw web-crawled audio into trainable speech-text data.

**Raw Audio.** We start with a large corpus (>10M hours) of conversational-speech audio crawled from the web. These are sourced from a range of web domains, filtered to remove other audio types like music, ads and background noise. Our audio corpus primarily consist of podcasts, interviews and monologue speeches.

**Speaker Diarization.** Our first processing step involves identifying different speakers in each audio sample. We use `pyannotate` (Bredin, 2023) to annotate each audio sample into speaker diarized outputs. For each audio, the diarization procedure outputs a list of (`audio-start`, `audio-end`, `speakerID`) triplets. An example of a diarization output on an audio sample is shown below:

```
[{'start': 0.031, 'end': 5.971, 'speaker': 'SPEAKER_06'},
 {'start': 7.085, 'end': 10.493, 'speaker': 'SPEAKER_06'},
 {'start': 11.607, 'end': 13.278, 'speaker': 'SPEAKER_06'},
 {'start': 13.565, 'end': 16.315, 'speaker': 'SPEAKER_06'},
 {'start': 17.092, 'end': 18.323, 'speaker': 'SPEAKER_06'},
 {'start': 25.968, 'end': 26.66,  'speaker': 'SPEAKER_01'}]
```

Here, the `start` and `end` markers denote the audio-timestamps corresponding to the beginning and end of the diarized segment, and the `speaker` denotes the speakerID corresponding to that segment. Note that there can be diarization segments with multiple overlapping timestamps, if the original audio has overlapping conversation.

**Language Filtering.** As a next step, we identify the primary language of the audio using Whisper (Radford et al., 2023) and filter out all non-english audio.

**Transcription Generation.** Next, we aim to provide paired text annotations for all of the raw audio in our corpus. For this, we first used the Whisper model (Radford et al., 2023) to transcribe the raw audio from each of the diarized output chunks. However, we noticed that the Whisper model transcriptions can tend to be quite noisy and contain some hallucinations. To ensure cleaner transcriptions, we use a post-processing transcription ensembling approach called ROVER (Fiscus, 1997) used in prior works performing transcription cleaning (Jalalvand et al., 2015). We first obtain additional speech transcriptions from an internal SIRI transcription model and `Nvidia-Parakeet-TDT-CTC`. We then apply the ROVER post-processing method using the three candidate transcriptions from Whisper, SIRI and Parakeet. We use the ensembled transcription as our text annotations for subsequent steps. We provide some examples of the individual model-based transcriptions and the final ROVER-ensembled transcriptions below:

> `Whisper:` " And I don't think it was a compliment. Yeah."
> `SIRI:` "And I don't think it as a compliment."
> `Parakeet:` "And I don't think it's compliment yeah."
> `ROVER-ensembled:` "And I don't think it was a compliment. Yeah. "
>
> `Whisper:` " Yeah, I was just never sure if it meant like someone who was left behind by fashion like..."
> `SIRI:` "Yeah, I was just never sure if it meant like someone who was left behind by fashion like"
> `Parakeet:` "Yeah, I was just never sure if it meant like someone who was left behind by fashion like"
> `ROVER-ensembled:` "Yeah, I was just never sure if it meant like someone who was left behind by fashion like "

**Transcription Filtering.** Despite the ROVER post-processing, we still find that a lot of annotations are low-quality including empty transcription texts and containing several repetitions. We filter out samples with such faulty transcriptions. For detecting repetition, we use a heuristic $n$-gram based approach. We first tokenize each transcription using a pretrained SentencePiece (Kudo & Richardson, 2018) tokenizer. We then search for unique 15-gram spans in the tokenized text. If we find that a 15-gram span occurs more than 5 times in the entire sequence, we discard that sample.

**Interleaved Chunking.** The last step in our pipeline is the interleaved chunking stage, which constructs the final audio-text chunks used for interleaved training. As described in the main text, we study two chunking strategies:

1. *Coarse interleaving.* Here, we aim to have relatively long audio-text chunks. To do this, we continually merge consecutive audio segments based on the diarization outputs while they have the same speakerID. While merging the segments, we concatenate the corresponding text transcriptions of each audio segment, separated by a white-space, to yield the merged text transcription for the merged audio.

2. *Fine interleaving.* Since the original diarized output segments already yield relatively short chunks, we do not apply any post-processing on the output segments and directly use them as our audio-text chunks for interleaved training.

For both chunking strategies, we additionally filter out any audio-text chunks where the audio chunk is shorter than 0.2 seconds.

## B    ETHICS STATEMENT

Our paper leverages large-scale web-crawled data for model pretraining. Below, we specify how our data collection complies with copyright, licensing, and other web-crawling policies. We further provide details on data provenance, consent, and compliance with legal and ethical standards (e.g., GDPR). For all other eval datasets, we use publicly open-sourced data and follow their respective usage policies.

1. **Data provenance.**  All speech data comes from publicly available podcast RSS feeds and similar spoken-word streams. We do not scrape behind paywalls and avoid clearly copyrighted catalogue content such as commercial audiobooks and music albums.

2. **Web-crawling policies.** Our collection framework respects `robots.txt` directives and website-specific terms of use.

3. **Licensing.** We preferentially include sources under permissive or podcast-typical licenses that allow redistribution for research. When license information is ambiguous, we err on the side of exclusion.

4. **Privacy and PII.** We apply automatic filters to reduce personally identifiable information (e.g., email addresses, phone numbers) and run safety classifiers to remove clearly harmful or sensitive content. We have used this data under our institution's data processing policies.

5. **GDPR and regional compliance.** Processing is conducted in accordance with internal legal guidance, and only aggregate, non-identifiable statistics are reported. No attempt is made to profile or target individual speakers.

## C    DETAILS OF SYNTHETIC DATASETS

### C.1    KNOWLEDGE-RICH DOMAINS USED FOR SYNTHETIC DATASETS

In this section, we provide a list of knowledge-rich domains we use for domain-filtering as the first step in our pipeline for constructing synthetic datasets:

1. https://www.numerade.com/home/

2. https://www.brainscape.com

3. https://brainly.com

4. https://www.chegg.com/

5. https://www.proprofs.com

6. https://www.schoolsolver.com

7. https://www.studypool.com

8. https://www.symbolab.com

9. https://www.justia.com

10. https://www.askalawyeroncall.com

11. https://freelawchat.com

12. https://www.healthtap.com

13. https://www.24houranswers.com

14. https://web2.0calc.com

15. https://myhomeworkapp.com

16. https://www.justanswer.com

17. https://quizlet.com/

### C.2    PROMPT

**Extraction prompt for *Krist*.** To extract and lightly rewrite the text content from the HTML using `gpt-4o-mini`, we use the following prompt:

> Extract the useful (non-boilerplate) text from the following HTML content into well-formatted plaintext, please. There is no need to retain hyperlinks out of the page, they can be dropped. Output the content in mark up tags as show below.
>
> ```
> ```plaintext
> {
> <well formatted plain text here>
> }
> {html_content}
> ```

**Question validation prompt for *Quest*.** To validate and filter out questions that are incorrectly formatted / extracted from the HTML, we use the following prompt to `gpt-4o`:

> Here is a problem that you do not need to solve:
> {question}
>
> ## Your task: Don't try to solve the problem, instead, do a brief free-form analysis, then output results for the following fields:
>
> complete: Values choose between (you can't use any other values)
> True - The problem is complete: it asks a clear and understandable question, and does not depend on any missing or unseen visual elements such as figures, graphs, tables, or images.
> False - The problem is incomplete: it is ambiguous, unanswerable, or relies on external content (e.g., a graph or diagram) that is not provided.
>
> is_question: Values choose between (you can't use any other values)
> True - The problem is asking a specific question (e.g., it requests the value of an expression, a numerical answer, or a specific outcome.
> False - The problem is not a question (e.g., it is a statement, conversation, or unrelated content).

**Question answering prompt for *Quest*.** Finally, we prompt gpt-4o to answer with a chain-of-thought to each verified question using:

> Please answer the following question. Let's think step by step.
>
> {question}

## C.3    EXAMPLES OF INVALID QUESTIONS IN QUEST

Previously, we presented the prompt used for validating and filtering out incomplete or incorrectly extracted questions. Since we score for *question completeness* and *question validity*, our filtering mechanism only keeps questions that are marked as complete *and* valid. Here, we show examples of questions that were marked as invalid i.e. marked as incomplete, invalid, or both.

> Question
> Example of mechanical?
> complete:  False
> is_question:  False
> ───────────────────────────────────
> Question
> How does this picture show social impacts of imperialism? helppp me
> complete:  False
> is_question:  True
> ───────────────────────────────────
> Question
> Minimum duration for diagnosis for: Selective Mutism
> complete:  True
> is_question:  False
> ───────────────────────────────────
> Question
> Audience analysis examples
> complete:  False
> is_question:  False
> ───────────────────────────────────

# D   TRAINING DATA STATISTICS

**Text-only dataset.** For our text-only continued pretraining dataset, we use the dataset used in the continual pretraining experiments of Li et al. (2025b), which roughly comprises of 2.2T tokens.

**Speech-text datasets.** Here, we provide the exact details of all our speech-text training data sources. Note that since our tokenizer processes audio at 12.5Hz, our token yield per second is 12.5 speech tokens. Hence, an hour of audio (3600s) corresponds to 45k speech tokens. In Tab. 8, for each dataset, we report the number of raw hours of speech content along with the total number of speech tokens. As is evident, web-crawl data contains the most number of unique tokens followed by Krist and Quest.

Table 8: **Training Data Statistics.**

| Training dataset | # Hours | # Speech tokens |
|---|---|---|
| Web-crawl | 8.03M | 361.3B |
| Krist | 4.72M | 212.4B |
| Quest | 0.86M | 38B |

## D.1   DETAILS OF DATA MIXTURES FOR SYNTHETIC DATA EXPERIMENTS

Here, we break down the exact token counts used for each data mixture in the experiments in Tab. 2. Remember that we train for a total of 200k steps with a batch-size of 512 and sequence-length of 16, 384 yielding 1.67T multimodal tokens for the full training run. For each experiment, we use 60% text-only and 40% speech-text mixing ratio. Hence, the text-only ratio corresponds to ∼1T tokens. The speech-text ratio corresponds to the remaining ∼670B tokens. Now, in Tab. 9, we report for each data source (text-only, web-crawl, Krist and Quest), the exact mixing proportion in the training mixture (%mix), total number of tokens in the training mixture (#toks) and the number of repeats (epochs) of the original data source (#repeats) used across all our experiments in Tab. 2. As is evident from the table, due to the heterogenity of data sources and their corresponding token-sizes, it is quite complex to determine an optimal mixing proportion. Our results also corroborate existing results in language (Guha et al., 2025) and vision-language (Bansal et al., 2025) reasoning domains, finding that mixing several data sources to improve performance is non-trivial.

Table 9: **Data mixture statistics for experiments in Tab. 2.**

| Training dataset | Text-only dataset | | | Web-crawl | | | Krist | | | Quest | | |
|---|---|---|---|---|---|---|---|---|---|---|---|---|
| | %mix | #toks | #repeats | %mix | #toks | #repeats | %mix | #toks | #repeats | %mix | #toks | #repeats |
| Web-crawl 100% | 0.60 | 1T | 0.45 | 0.40 | 670B | 1.85 | 0.00 | 0.00 | 0.00 | 0.00 | 0.00 | 0.00 |
| Web-crawl 53% + Krist 47% | 0.60 | 1T | 0.45 | 0.21 | 355B | 0.98 | 0.19 | 315B | 1.48 | 0.00 | 0.00 | 0.00 |
| Web-crawl 66% + Quest 34% | 0.60 | 1T | 0.45 | 0.26 | 442B | 1.22 | 0.00 | 0.00 | 0.00 | 0.14 | 228 | 6.00 |
| Web-crawl 59% + Quest 6% + Krist 35% | 0.60 | 1T | 0.45 | 0.24 | 395B | 1.09 | 0.14 | 232B | 1.10 | 0.02 | 43B | 1.13 |
| Web-crawl 40% + Quest 27% + Krist 33% | 0.60 | 1T | 0.45 | 0.16 | 267B | 0.74 | 0.13 | 221B | 1.04 | 0.11 | 182 | 4.79 |

# E   TRAINING DETAILS

All our models are $3.8$B-parameter transformer-based (Vaswani et al., 2017) speech-language models. We use a global-batch-size of $512$ for all our experiments. Our models use a packed-sequence-length of $16,384$ tokens. We train for 200k steps in total, yielding a total of $1.67$T multimodal tokens for our training runs. Using the standard $6ND$ rule (Kaplan et al., 2020), this equates to about $3.81\times10^{22}$FLOPs (note that this estimate is a rough lower bound since we do not count the FLOPs associated with the speech tokenizer in this estimate). We only tune the language model weights while keep the speech tokenizer frozen. We use a cosine-decay learning rate schedule with $1000$ steps of linear-warmup. We use the AdamW (Loshchilov & Hutter, 2017) optimizer with $\beta_1{=}0.9$ and $\beta_2{=}0.95$, a peak learning rate of $3e{-}4$, weight decay of $1e{-}5$ and clip gradients to a max norm of $1.0$. We use the `axlearn` (Lee et al., 2025) codebase for all our experiments using `jax` (Bradbury et al., 2021) and `pygrain` (Ritter et al., 2023) for dataloading. One training run takes approximately $7$ days on $512$ TPU-v6e chips.

# F EXTENDED RELATED WORK

In the main paper, we briefly described some related work in speech-language pretraining. Further, we focused on situating our work in the SpeechLM literature and emphasized the lack of data-centric research in speech-language pretraining. Here, we provide a deeper dive into SpeechLMs and reference some related data-centric work that does exist in the speech-language domain.

**Speech Language Models.** There has been a recent push for training end-to-end SpeechLMs (Arora et al., 2025). Early efforts like Whisper (Radford et al., 2023), SALMONN (Tang et al., 2023), and LTU-AS (Gong et al., 2023) employed multi-task pretraining to enable tasks like automatic speech recognition, emotion classification etc. Scaling these principles by increasing model-size and training compute (Chu et al., 2023; 2024; Liu et al., 2025; Geng et al., 2025; Kong et al., 2024; Ghosh et al., 2025; Goel et al., 2025) has yielded continued gains. Further works considered pretraining models with speech understanding and generation capabilities (Lakhotia et al., 2021; Algayres et al., 2023; Hassid et al., 2023; Nguyen et al., 2025b; Nachmani et al., 2023; Rubenstein et al., 2023; Zhang et al., 2023; Défossez et al., 2024). More recently, models like Kimi-Audio (Ding et al., 2025), Step-Audio-2 (Wu et al., 2025a), Baichuan-Audio (Li et al., 2025c), GLM-4-Voice (Zeng et al., 2024a), and MiMo-Audio (Xiaomi, 2025) have emerged as strong foundation models that seamlessly perform several tasks, including spoken-question answering. While demonstrating impressive performance, details behind their data curation strategies are scant. Through our controlled experiments, we aim to fill this gap by shedding light on how to effectively construct speech-text pretraining datasets.

**Data Curation for Speech-Language Models.** Whisper (Radford et al., 2023) was one of the first works to effectively leverage web-scale data for training a multi-task speech-text model, using a dataset of 680k hours. Attempting to openly reproduce the original Whisper dataset, (Ngo et al., 2025) introduced OLMoASR-POOL, a dataset of 3M hours of audio and 17M transcripts. They conducted heuristic-based filtering on their data pool, showcasing benefits on ASR tasks. Tian et al. (2024) and Peng et al. (2025) similarly conducted comprehensive studies to understand the effects of data heterogenity, ASR error rate based filtering and LLM-based transcription rephrasing, while training Whisper-style models. However, these efforts were limited to training models that were primarily capable of performing ASR tasks. The data curation literature in the end-to-end SpeechLM literature is much more sparse. Kimi-Audio (Ding et al., 2025) describes their speech-text dataset construction pipeline, beginning from 13M audio hours and processing them into speech-text interleaved training data. However, why certain design decisions were taken remain unanswered. Contrarily, Zeng et al. (2024b) constructed synthetic interleaved data sourced from high-quality text pretraining data, but yet again omit clear details on key design choices. MiMo-Audio (Xiaomi, 2025) scaled up their training dataset size by an order of magnitude to an unprecedented 100M hours of audio data. While they showcased the benefits of dataset quantity using few-shot experiments, they did not conduct any explicit controlled experiments to justify the filtering and curation decisions they made. In our work, we aim to fill this gap on the data-centric side of SpeechLMs, by describing and understanding data curation pipelines for speech-text interleaved pretraining through three key questions around interleaved data chunking, synthetic dataset construction and modality sampling schemes during interleaved training.

## G    DETAILS AND EXAMPLES OF SQA EVALUATION DATASETS

We aim to evaluate the *speech-to-text transfer* capability of SpeechLMs, where the model is asked a question in speech and tasked with responding in text (S→T). In the literature, there is a lack of standardized evaluations for this task of Spoken-Question-Answering (SQA). While efforts like Spectron-LM (Nachmani et al., 2023) and Voxtral (Liu et al., 2025) have open-sourced some evaluation sets, they use different text-to-speech engines and generation parameters for synthesizing the spoken questions, rendering comparisons across different models unfair. Moreover, these datasets only consist of a question and answer, requiring models to generate free-form text outputs. However, prior works in LM evaluation standardization (Gu et al., 2024; Allal et al., 2025; Li et al., 2024; Brown et al., 2020) recommend using a *cloze-form* of MCQ evaluation for evaluating base-models with question-conditioned completion log-probabilities rather than decoding free-form text outputs. The log-probability method removes evaluation confounds such as decoding temperature, sampling method and other decoding parameters, which are known to induce large variance (Hochlehnert et al., 2025). Therefore, we construct a standardized SQA evaluation suite of three datasets—*Spoken-LLaMA-Questions*, *Spoken-Web-Questions* and *Spoken-TriviaQA*. We source the raw audio questions from OpenAudioBench (Li et al., 2025c). We then prompt `gpt-4o-mini` with the original text question and answer of each sample to provide a set of three distractor choices (the prompts for generating choices are in Appx. H). Hence, our final evaluation datasets consist of a spoken-question and 4 choices, with one correct answer (chance-level is 25%). In Tab. 10, we provide details about the number of test samples, the TTS engine used for synthesizing the speech questions, and the links to the original audio source files.

Table 10: **Details of SQA evaluation datasets.**

| Evaluation Dataset | Num. samples | Chance% | TTS Engine | Audio Source |
|---|---|---|---|---|
| Spoken-LLaMA-Questions | 300 | 25% | Google Cloud TTS | Link |
| Spoken-TriviaQA | 1000 | 25% | Baichuan-Audio TTS | Link |
| Spoken-Web-Questions | 1000 | 25% | Baichuan-Audio TTS | Link |

Below, we also provide a few examples from each evaluation dataset, with the question (in text), choices, and the ground-truth answer.

- *Spoken-LLaMA-Questions*

> `Question`: What is the capital of France?
> `Choices`: Paris, London, Berlin, Madrid
> `Ground-Truth`: Paris
>
> `Question`: Which river is the longest in South America?
> `Choices`: Nile, Amazon, Paraná, Orinoco
> `Ground-Truth`: Amazon

- *Spoken-TriviaQA*

> `Question`: Who was Jackie Kennedy's second husband?
> `Choices`: John F. Kennedy, Robert F. Kennedy, Frank Sinatra, Aristotle Onassis
> `Ground-Truth`: Aristotle Onassis
>
> `Question`: What is the oldest vegetable known to man?
> `Choices`: Carrot, Potato, Pea, Onion
> `Ground-Truth`: Pea

- *Spoken-Web-Questions*

> `Question:` What language do most Italians speak?
> `Choices:` Italian, French, Spanish, German
> `Ground-Truth:` Italian
>
> `Question:` Who did Shaq first play for?
> `Choices:` Los Angeles Lakers, Miami Heat, Boston Celtics, Orlando Magic
> `Ground-Truth:` Orlando Magic

**Evaluation details.** We use log-likelihood based scoring for our evaluation protocol following standard language modeling works (Brown et al., 2020; Allal et al., 2025; Gu et al., 2024).

For each test sample and each answer-choice (out of 4 total choices), we use the following cloze-form to prompt the model:

```
Question:\n<question-in-audio>\nAnswer:<answer-choice>
```

Then, we compute the completion log-probability for each of the 4 answer choices. We normalize the completion log-probability by answer length to prevent biasing against long answer choices. A question is marked correct if the model assigns highest normalized log-probability to the ground-truth answer. We use standard accuracy metric (random chance level is 25%) for reporting results. For running all our model evaluations, we use a fork of `lm-eval-harness` (Gao et al., 2024a).

## H   PROMPTS FOR GENERATING DISTRACTOR CHOICES FOR EVALUATION SETS

We use the following prompt for generating the distractor options for *Spoken-LLaMA-Questions* and *Spoken-TriviaQA*.

```
SYSTEM PROMPT
```
You are a helpful assistant.

```
INPUT PROMPT
```
I will give you a simple question and answer pair. This pair comes from an evaluation dataset. I am trying to convert it into an MCQ format dataset. You have to give three more plausible distractor options that I can use along with the correct option to create the MCQ test set. Give the three distractor options one after the other, comma-separated, all in one line.

Here are a few examples:

Input:
Question: What colour is the sky?
Answer: blue

Output:
green,red,yellow

Input:
Question: What season comes after spring?
Answer: summer

Output:
winter,monsoon,autumn

I will give you the question and the answer now. Remember, please give the three options in one line, comma-separated.

Question: `<question>`
Answer: `<answer>`

For *Spoken-Web-Questions*, as there can be multiple correct answers for a question, we pick the first reference answer as ground-truth and use the following prompt for generating distractor options.

SYSTEM PROMPT
You are a helpful assistant.

INPUT PROMPT
I will give you a simple question and answer pair. This pair comes from an evaluation dataset. Note that the answer might be one of out many possible correct answers. I am trying to convert it into an MCQ format dataset. You have to give three more plausible distractor options that I can use along with the correct option to create the MCQ test set. Since the provided answer might be one of many possible correct answers, ensure that the distractor options you provide are definitely incorrect for the given question. For example, if the question is "What is a leap year?" and the answer I provide is 2004, do not give distractor options like 2000 or 2012. Give the three distractor options one after the other, comma-separated, all in one line.

Here are a few examples:

Input:
Question: What colour is the sky?
Answer: blue

Output:
green,red,yellow

Input:
Question: What season comes after spring?
Answer: summer

Output:
winter,monsoon,autumn

I will give you the question and the answer now. Remember, please give the three options in one line, comma-separated.

Question: <question>
Answer: <answer>

# I PROMPT TEMPLATE FOR GPT-4O-AUDIO IN AUTO EVAL

We use the following prompt template when using `GPT-4o-audio` in our auto evaluation pipeline for audio responses.

---

SYSTEM PROMPT

Please act as an impartial judge and evaluate the quality of the responses provided by two AI assistants.

INPUT PROMPT:

¡user_audio¿

You are given an audio clip from a user talking to an AI assistant. And you will be given two audio responses to this user request. The first response is denoted as *Response A* and the second response is denoted as *Response B*. Your job is to evaluate which response is better.

Begin your evaluation by first generating your own answer to the user's request. You must provide your answers before judging any answers.

Here is the transcript of the audio clip to help you understand the conversation history: ¡user_audio_transcription¿.

When evaluating the responses, compare both responses with your answer. You must identify and correct any mistakes or inaccurate information.

Then consider if the responses are helpful, relevant, and concise. Helpful means the answer correctly responds to the prompt or follows the instructions. Note when user request has any ambiguity or more than one interpretation, it is more helpful and appropriate to ask for clarifications or more information from the user than providing an answer based on assumptions. Relevant means all parts of the response closely connect or are appropriate to what is being asked. Concise means the response is clear and not verbose or excessive.

Then consider if the responses correctly understand user's emotion and address user's request in a considerate, empathetic, and appropriate manner.

Then consider the creativity and novelty of the responses when needed.

Finally, identify any missing important information in the responses that would be beneficial to include when responding to the user request.
After providing your explanation, you must output only one of the following choices as your final verdict:
1. Response A is better: [[A>B]]
2. Response B is better: [[B>A]]
3. Tie, relatively the same: [[A=B]]

---

## J  DIVERGENCE ANALYSIS BETWEEN MODALITY DISTRIBUTIONS

In this section, we describe in detail the exact setup used for our analysis in Sec. 4.1.

We start with a spoken question-answering test set. Each test sample consists of $(q_a, q_t, gt)$ triplets, where $q_a$ denotes the spoken question in audio modality, $q_t$ denotes the question in text modality, and $gt$ denotes the ground-truth answer in text modality.

**Goal.**  We aim to measure the divergence between the token-wise teacher-forced (Williams & Zipser, 1989) conditional probability distributions of the audio and text modality. That is, we compare the next–token distributions under audio vs. text question conditioning, evaluated along the same ground–truth (GT) answer path (the answer is always in text modality).

**Notation.**  For each test sample $s$, let $\{t_1, t_2 \cdots t_m\}$ and $\{a_1, a_2 \cdots a_n\}$ represent the question tokens in text and audio modality respectively. That is, the tokenized representation of $q_t$ is $\{t_1, t_2 \cdots t_m\}$ and the tokenized representation of $q_a$ is $\{a_1, a_2 \cdots a_n\}$. For brevity, let us denote these tokenized representations as $t_{1:m}$ and $a_{1:n}$. Note that since the length of the question tokens in text and audio modalities might differ, it is possible that $n \neq m$. Let $\{g_1, g_2 \cdots g_o\}$ represent the ground-truth answer tokens in text modality i.e. the tokenized representation of $gt$ is $\{g_1, g_2 \cdots g_o\}$. Again, for brevity, we denote this as $g_{1:o}$. Let $V$ be the vocabulary of the SpeechLM.

For a given test sample $s$, for each answer token $i \in \{1, 2 \cdots o\}$, we define the teacher-forced next–token distributions as:

$$P_{\text{aud},i}^{(s)}(v) = \Pr_{\theta}\big(X = v \mid a_{1:n}, g_{1:i-1}\big), \quad v \in V, \tag{1}$$

$$P_{\text{text},i}^{(s)}(v) = \Pr_{\theta}\big(X = v \mid t_{1:m}, g_{1:i-1}\big), \quad v \in V. \tag{2}$$

where $\Pr_{\theta}(X{=}v|Y)$ represents the conditional probability distribution for all values $v \in V$, conditioned on the previous context $Y$.

**Per–token divergences.**  We now compute (1) forward KL, (2) reverse KL, and (3) Jensen–Shannon (JS) divergence at each step $i$, between the two next-token distributions:

$$D_{\text{KL}\rightarrow}^{(s)}(i) = \sum_{v \in V} P_{\text{aud},i}^{(s)}(v) \, \log \frac{P_{\text{aud},i}^{(s)}(v)}{P_{\text{text},i}^{(s)}(v)}, \tag{3}$$

$$D_{\text{KL}\leftarrow}^{(s)}(i) = \sum_{v \in V} P_{\text{text},i}^{(s)}(v) \, \log \frac{P_{\text{text},i}^{(s)}(v)}{P_{\text{aud},i}^{(s)}(v)}, \tag{4}$$

$$D_{\text{JS}}^{(s)}(i) = \tfrac{1}{2} D_{\text{KL}}\big(P_{\text{aud},i}^{(s)} \,\big\|\, M_i^{(s)}\big) + \tfrac{1}{2} D_{\text{KL}}\big(P_{\text{text},i}^{(s)} \,\big\|\, M_i^{(s)}\big), \, M_i^{(s)} = \tfrac{1}{2}\Big(P_{\text{aud},i}^{(s)} + P_{\text{text},i}^{(s)}\Big). \tag{5}$$

**Answer–span aggregation (per example).**  To get a mean divergence value per sample, we average the per-token divergences over the answer length $o$ (masking any padded positions in practice):

$$\overline{D}_{\text{KL}\rightarrow}^{(s)} = \frac{1}{o} \sum_{i=1}^{o} D_{\text{KL}\rightarrow}^{(s)}(i), \qquad \overline{D}_{\text{KL}\leftarrow}^{(s)} = \frac{1}{o} \sum_{i=1}^{o} D_{\text{KL}\leftarrow}^{(s)}(i), \qquad \overline{D}_{\text{JS}}^{(s)} = \frac{1}{o} \sum_{i=1}^{o} D_{\text{JS}}^{(s)}(i). \tag{6}$$

The distribution of these per-sample mean divergences is what we plot in Fig. 5 and Appx. J.1.

**Dataset–level metrics.**  Over each test set $S$ we also report the dataset means across metrics in Tab. 11:

$$\mathcal{D}_{\text{KL}\rightarrow} = \frac{1}{|S|} \sum_{s \in S} \overline{D}_{\text{KL}\rightarrow}^{(s)}, \qquad \mathcal{D}_{\text{KL}\leftarrow} = \frac{1}{|S|} \sum_{s \in S} \overline{D}_{\text{KL}\leftarrow}^{(s)}, \qquad \mathcal{D}_{\text{JS}} = \frac{1}{|S|} \sum_{s \in S} \overline{D}_{\text{JS}}^{(s)}. \tag{7}$$

### J.1 MORE RESULTS ACROSS DIFFERENT METRICS AND TEST SETS

In the main paper Sec. 4.1, we showcased the divergence plots between the conditional next-token distributions, on the Spoken-LLaMA-Questions test with the reverse KL-divergence metric only. Here, we showcase the divergence distributions across all three of our test sets—Spoken-LLaMA-Questions, Spoken-Web-Questions and Spoken-TriviaQA—across three divergence metrics—Forward KL Divergence, Reverse KL Divergence and Jensen Shannon Divergence. The plots for Spoken-LLaMA-Questions are in Fig. 10, for Spoken-Web-Questions are in Fig. 11, and for Spoken-TriviaQA are in Fig. 12. Furthermore, in Tab. 11, we report the mean values of the divergence distributions obtained. Across all plots and the table, we observe that our data interventions consistently close the distribution mismatch between the conditional probability distributions of audio and text modalities. This suggests that our data intervention implicitly induce a self-distillation behaviour (Zhang et al., 2021a; Mobahi et al., 2020; Zhang et al., 2019) in our trained SpeechLMs. Such an implicit "distillation through data" property has also been observed in prior works in the multimodal and language domains (Udandarao et al., 2025; Rawat et al., 2024; Wang et al., 2024; Sachdeva & McAuley, 2023; Wang et al., 2018). Further, Wang et al. (2025a) showed that explicitly applying a cross-modal distillation objective further helps to reduce the modality distribution gap, and our results further implicitly confirm this. In the future, further methods that have been proposed to reduce the modality gap in vision-language models (Schrodi et al., 2024; Udandarao, 2022; Liang et al., 2022; Li et al., 2025a) can also be experimented with in the speech-language domain.

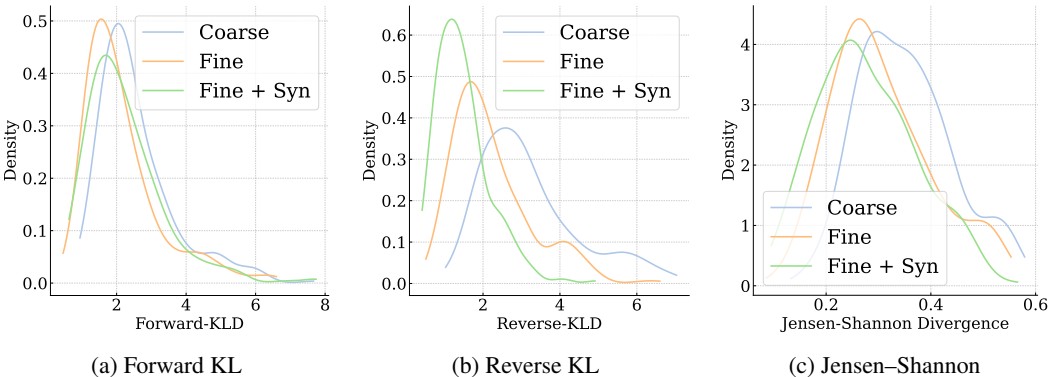

Figure 10: **Conditional-distribution divergences on Spoken-LLaMA-Questions**.

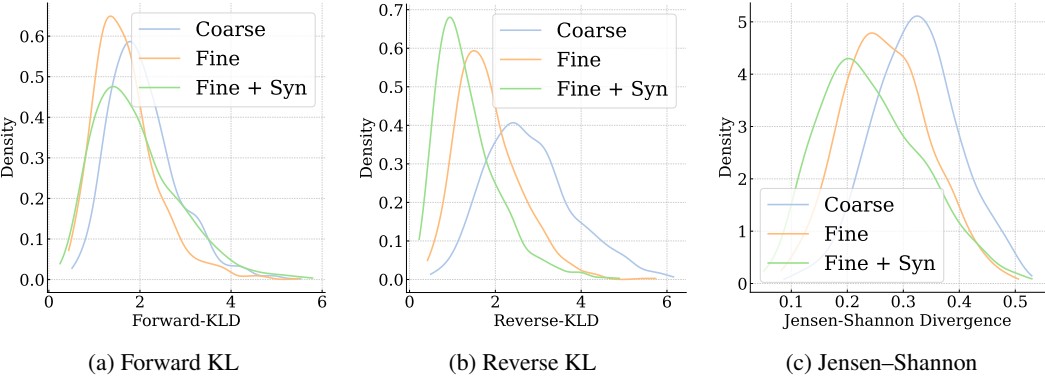

Figure 11: **Conditional-distribution divergences on Spoken-Web-Questions**.

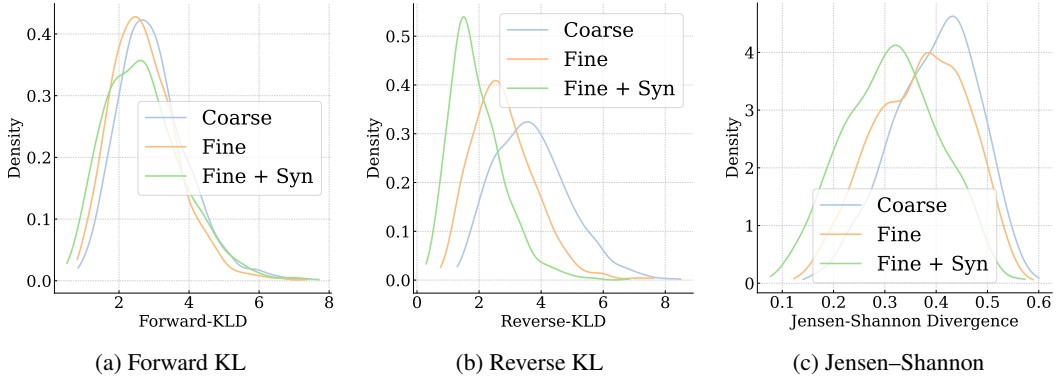

Figure 12: **Conditional-distribution divergences on Spoken-TriviaQA**.

Table 11: **Dataset-level means of all divergence metrics b/w conditional next-token distributions.** We report the means of all three divergence distributions (as computed in eq. (7)). *FKL* represents forward KL-divergence, *RKL* is reverse KL-divergence and *JSD* is Jensen-Shannon divergence.

| Method | Spoken-Web-Questions | | | Spoken-TriviaQA | | | Spoken-LLaMA-Questions | | |
|---|---|---|---|---|---|---|---|---|---|
| | *FKL* | *RKL* | *JSD* | *FKL* | *RKL* | *JSD* | *FKL* | *RKL* | *JSD* |
| Coarse | 2.07 | 2.78 | 0.32 | 2.97 | 3.70 | 0.40 | 2.57 | 3.20 | 0.35 |
| Fine | 1.68 | 1.84 | 0.27 | 2.72 | 2.80 | 0.36 | 2.15 | 2.21 | 0.30 |
| Fine + Syn | 1.90 | 1.35 | 0.24 | 2.71 | 1.94 | 0.31 | 2.23 | 1.47 | 0.27 |

## K  TOPIC DOMAIN ANALYSIS

### K.1  DETAILS ABOUT TOPIC DOMAIN CLASSIFIER

For conducting the topic domain analysis in Fig. 6, we used the topic domain classifier that was released by (Wettig et al., 2025). The classifier is a `gte-base-en-v1.5` model that was fine-tuned on web-texts annotated by LLaMA models. We used the `No-URL` version of the classifier that takes only the raw text as input and classifies it into one of 24 output classes. For getting the topic distribution of each of our datasets, we randomly sample 5000 examples, concatenate all the text chunks from each example (for web-crawled data, these are the annotated transcriptions while for synthetic data, these are the source text data samples), and use that as input to the topic classifier.

### K.2  TOPIC DISTRIBUTION FOR SPOKEN-WEB-QUESTIONS

In Fig. 13, we showcase the topic distribution of Spoken-Web-Questions. Similar to the takeaways in Fig. 6, we find that some of the topics that Spoken-Web-Questions contains are severely under-represented in the web-crawled dataset while being represented adequately in the synthetic datasets. This further corroborates our findings that synthetic datasets help close the distribution mismatch between the web-crawled dataset and the evaluation datasets. Our findings regarding the under-representation of concepts in web-crawled datasets have also been echoed in the language and vision domains (Wiedemer et al., 2025; Parashar et al., 2024; Elazar et al., 2023; Kandpal et al., 2023; Udandarao et al., 2024; Zhao et al., 2024; Samuel et al., 2024; Dodge et al., 2021).

### K.3  A MORE FINE-GRAINED TOPIC DISTRIBUTION ANALYSIS

For all the topic domain analyses we have conducted previously, we used a coarse-level topic classifier that could categorize between 24 different topics. Here, we use a more fine-grained topic classifier that can produce a finer-grained categorization into 67 different topics. We use the `finefineweb-domain-fasttext-classifier`, which is a bi-gram fasttext model that was used for curating the FineFineWeb dataset (Zhang et al., 2024a). We use the same procedure as

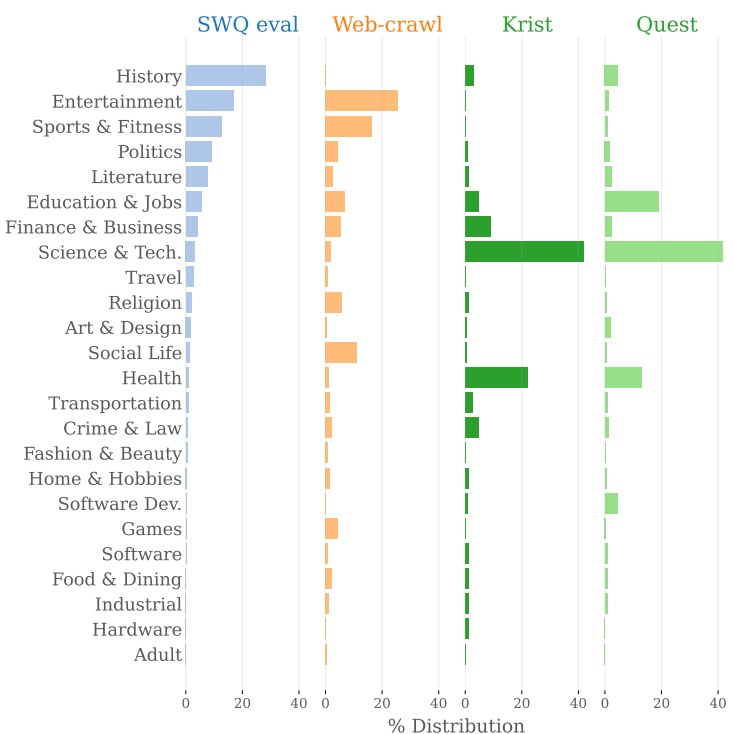

Figure 13: Topic domain distribution for *Spoken-Web-Questions* eval and training datasets.

before for annotating our evaluation and training datasets. We plot the fine-grained topic distributions for *Spoken-LLaMA-Questions* in Fig. 14, *Spoken-TriviaQA* in Fig. 15 and *Spoken-Web-Questions* in Fig. 16, along with all training datasets. Across all the plots, our findings from Figs. 6 and 13 hold—our synthetic datasets increase the diversity and topic coverage of our training data distribution, thereby more closely matching the distribution of concepts encompassed in the evaluation datasets. This helps improve model generalization, yielding better downstream performance.

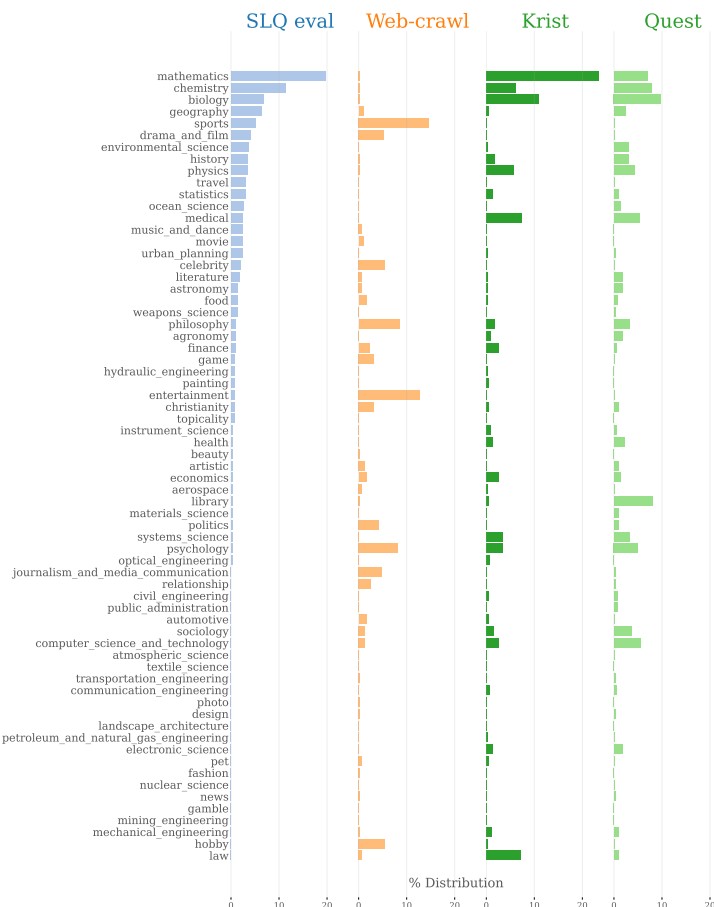

Figure 14: Fine-grained topic domain distribution for *Spoken-LLaMA-Questions* eval and training datasets.

## L   DETAILS ABOUT CONTAMINATION ANALYSIS

### L.1   EXAMPLES OF CONTAMINATED MATCHES

In this section, we show some examples of the matches we get from our contamination identification procedure. For each match, we show the training dataset, the training sample, the contaminated test sample, the test dataset it belongs to, and the contaminated $n$-gram span.

```
Train dataset: Quest
Train sample:
```
What is the definition of vitreous? The word derives from Latin viteus, "of glass," and is used to describe either a glass-like quality or the glass-like substance filling the eye. Vitreous (adjective): 1. Having the appearance or properties of glass; glassy, transparent, brittle. 2. In anatomy, relating to the vitreous humor or vitreous body—the clear, gelatinous substance filling the space between the lens and the retina of the eye.
```
Test dataset: Spoken-TriviaQA
Test sample:
```
What is the thick watery substance filling the space between the lens and the retina of the eye?
```
Contaminated span:
```
substance filling the space between the lens and the retina of the eye

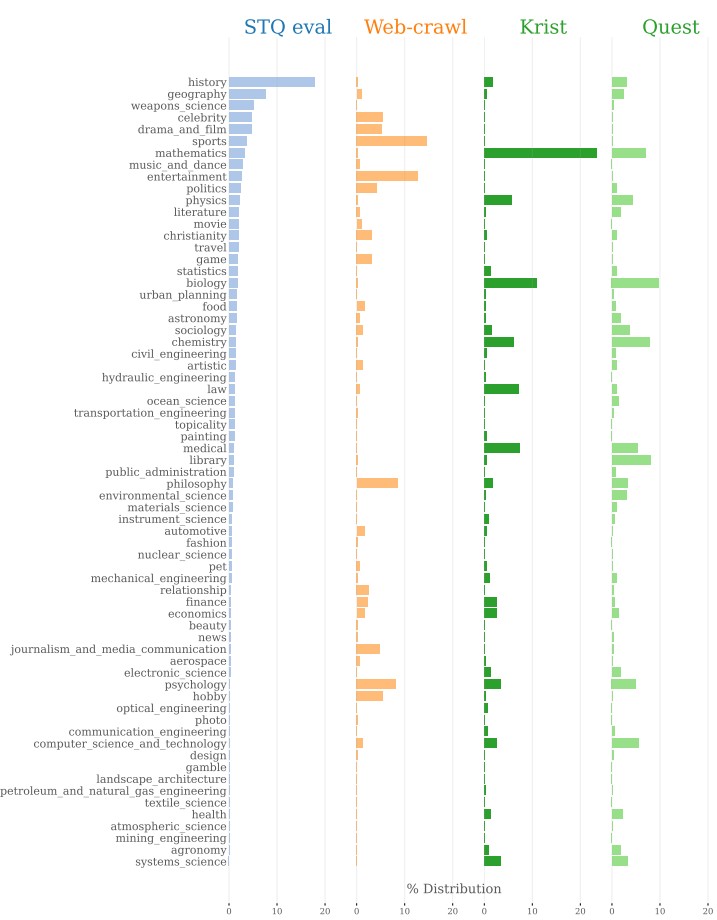

Figure 15: Fine-grained topic domain distribution for *Spoken-TriviaQA* eval and training datasets.

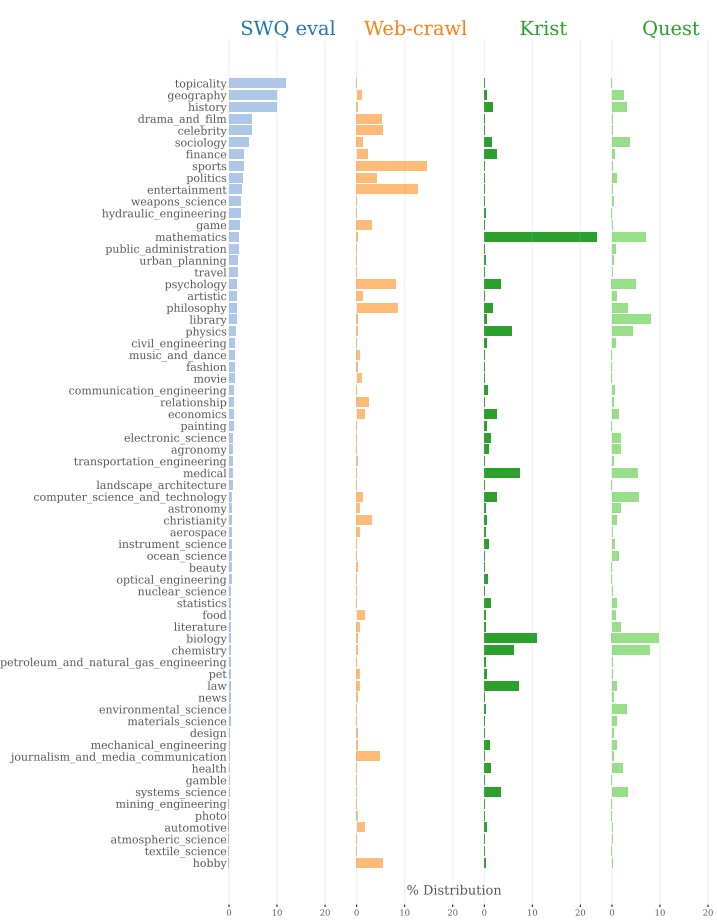

Figure 16: Fine-grained topic domain distribution for *Spoken-Web-Questions* eval and training datasets.

```
Train dataset: Quest
Train sample:
```
When did Arthur & Catherine marry? Prince Arthur, the eldest son of Henry VII, married Catherine of Aragon — daughter of Ferdinand II of Aragon and Isabella I of Castile. Their wedding took place at St. Paul's Cathedral in London on 14 November 1501. Arthur and Catherine were married on 14 November 1501.
```
Test dataset: Spoken-TriviaQA
Test sample:
```
What was founded by Ferdinand II of Aragon and Isabella I of Castile to keep Catholic orthodoxy as the major religion of their kingdoms?
```
Contaminated span:
```
ferdinand ii of aragon and isabella i of castile

```
Train dataset: Krist
Train sample:
```
What conclusions can be drawn about the USA's actions in the 1920s and 1930s? One conclusion to this statement, which seems to be addressing the approach to foreign policy during the period, might be "...reflected a strong, if uneven, commitment to isolationism." On the one hand, the United States was fairly steadfast in its unwillingness to get directly involved in the affairs of the world, particularly Europe. Except for a few non-binding pacts and negotiations over the repayment of reparations and war debts, the United States remained generally aloof from European affairs during the 1920s.
```
Test dataset: Spoken-TriviaQA
Test sample:
```
What was the name of the democratic government of Germany in the 1920s and early 1930s, destroyed by Adolf Hitler?
```
Contaminated span:
```
in the 1920s and early 1930s,

```
Train dataset: Krist
Train sample:
```
In 1912, Lenin, then in exile in Switzerland, appointed Joseph Stalin to serve on the first Central Committee of the Bolshevik Party. Three years later, in November 1917, the Bolsheviks seized power in Russia. The Soviet Union was founded in 1922, with Lenin as its first leader. During these years, Stalin had continued to move up the party ladder, and in 1922 he became secretary general of the Central Committee of the Communist Party, a role that enabled him to appoint his allies to government jobs and grow a base of political support.
```
Test dataset: Spoken-Web-Questions
Test sample:
```
what led to stalin rise in power?
```
Contaminated span:
```
to serve on the first central committee of the bolshevik party
```

```
Train dataset: Krist
Train sample:
James Harold Doolittle
Doolittle, James Harold (1896– ), U.S. pilot and World War II air hero. Famous as a racing
pilot in the 1920s and early 1930s, he led the first air raid on Tokyo on April 18, 1942, thereby
slowing the Japanese offensive. After the war he was an executive in the aerospace industry.
See also: World War II.
Test dataset: Spoken-TriviaQA
Test sample:
What was the name of the democratic government of Germany in the 1920s and early 1930s,
destroyed by Adolf Hitler?
Contaminated span:
in the 1920s and early 1930s,
```

## L.2 Proportion of contamination in eval datasets

Table 12: **Proportion of contamination.** For each evaluation dataset, we report the proportion of test samples detected as contaminated. We also report the absolute number of matches in brackets.

| Evaluation dataset | % Contamination [# samples] | | |
|---|---|---|---|
| | *Krist* | *Quest* | *All* |
| Spoken-Web-Questions | 0.4% [4] | 0.1% [1] | 0.4% [4] |
| Spoken-TriviaQA | 2.2% [22] | 0.8% [8] | 2.5% [25] |
| Spoken-LLaMA-Questions | 6.7% [20] | 0.2% [5] | 7.7% [23] |

## L.3 Expanded description of significance testing setup and results

**Null hypothesis.** We start from the full test set (containing contaminated samples). In our significance test, we test *whether removing contaminated test items reduces accuracy beyond what would be expected under random removal of an equal number of items.* Formally, for accuracy $A$, the null is:

$$H_0 : A_{\text{clean}} \sim \text{distribution of } A_{\text{rand}},$$

i.e., the clean accuracy is not lower than the random-removal distribution. Because the contamination claim is directional (contamination would inflate accuracy), we use a *one-sided* test.

**Test procedure.** For each training mix and dataset from Sec. 3.4, we compute: (i) *Full* accuracy on the full test set; (ii) *Clean* accuracy after removing all known contaminated items; (iii) a *random-removal baseline* by drawing 100 random subsets (without replacement) of the same size as the contaminated set, recomputing accuracy on the remaining items each time. Accuracies for (ii) and (iii) are computed over the reduced denominators (remaining items). From the bootstrap distribution we report the mean and 95% percentile CI and compute the empirical one-sided $p$-value as:

$$p = \Pr\big(A_{\text{rand}} \leq A_{\text{clean}}\big),$$

This $p$-value is appropriate for the hypothesis that contamination inflates accuracy (so clean should be lower if inflation is present). With 100 replicates, the $p$-value granularity is 0.01. Hence, we report $p<0.01$ when no replicate from the bootstrap distribution is as low as the clean accuracy.

**Results and interpretation.** Tables 13–15 summarize results for Spoken-TriviaQA, Spoken-LLaMA-Questions, and Spoken-Web-Questions. We highlight the difference $\Delta = \text{Clean}-\text{RandMean}$ and give the decision at a significance level $\alpha=0.01$.

**Takeaways.** Across *STQ* and *SWQ*, clean accuracies consistently fall within the random-removal confidence intervals. Therefore, *we find no significant contamination-driven inflation.* For *SLQ*, the Web-crawl 59% + Quest 6% + Krist 35% mix shows a drop in clean accuracy relative to the

Table 13: One-sided contamination test on STQ (N=1000).

| Data mix | Full (%) | Clean (%) | Random mean (95% CI) (%) | $\Delta$ (pp) | One-sided $p$ | Decision |
|---|---|---|---|---|---|---|
| Web-crawl 53% + Krist 47% | 29.20 | 29.03 | 29.22 [28.77, 29.64] | −0.19 | 0.32 | Fail to reject $H_0$ |
| Web-crawl 66% + Quest 34% | 34.70 | 34.56 | 34.73 [34.26, 35.28] | −0.17 | 0.38 | Fail to reject $H_0$ |
| Web-crawl 59% + Quest 6% + Krist 35% | 30.80 | 30.46 | 30.81 [30.36, 31.18] | −0.35 | 0.09 | Fail to reject $H_0$ |
| Web-crawl 40% + Quest 27% + Krist 33% | 31.70 | 31.59 | 31.70 [31.28, 32.10] | −0.11 | 0.41 | Fail to reject $H_0$ |

Table 14: One-sided contamination test on SLQ (N=300).

| Training mix | Full (%) | Clean (%) | Random mean (95% CI) (%) | $\Delta$ (pp) | One-sided $p$ | Decision |
|---|---|---|---|---|---|---|
| Web-crawl 53% + Krist 47% | 52.00 | 50.54 | 52.16 [50.54, 53.62] | −1.62 | 0.10 | Fail to reject $H_0$ |
| Web-crawl 66% + Quest 34% | 66.33 | 66.79 | 66.34 [64.62, 68.06] | +0.45 | 0.82 | Fail to reject $H_0$ |
| Web-crawl 59% + Quest 6% + Krist 35% | 50.33 | 48.01 | 50.33 [48.91, 51.62] | −2.32 | < 0.01 | **Reject** $H_0$ |
| Web-crawl 40% + Quest 27% + Krist 33% | 49.33 | 47.29 | 49.43 [47.65, 50.90] | −2.14 | 0.02 | Fail to reject $H_0$ |

Table 15: One-sided contamination test on SWQ (N=1000).

| Training mix | Full (%) | Clean (%) | Random mean (95% CI) (%) | $\Delta$ (pp) | One-sided $p$ | Decision |
|---|---|---|---|---|---|---|
| Web-crawl 53% + Krist 47% | 43.40 | 43.27 | 43.39 [43.22, 43.57] | −0.12 | 0.23 | Fail to reject $H_0$ |
| Web-crawl 66% + Quest 34% | 42.70 | 42.57 | 42.69 [42.47, 42.87] | −0.12 | 0.20 | Fail to reject $H_0$ |
| Web-crawl 59% + Quest 6% + Krist 35% | 43.80 | 43.67 | 43.80 [43.62, 43.98] | −0.13 | 0.19 | Fail to reject $H_0$ |
| Web-crawl 40% + Quest 27% + Krist 33% | 43.30 | 43.17 | 43.29 [43.07, 43.47] | −0.12 | 0.23 | Fail to reject $H_0$ |

random baseline that is statistically significant under our one-sided $p$-test ($p<0.01$), consistent with contamination inflating test performance. However, for the other three data mixes we again see no significant evidence of inflation, under our testing setup. Hence, overall we conclude that *contamination does not have a major effect* on inflating model performance.

## L.4 LIMITATIONS OF OUR CONTAMINATION ANALYSIS

**Post-hoc analysis.** Our contamination analysis is entirely post-hoc, after training of a model is complete. In the ideal case, one would decontaminate the training sets with respect to the test sets a-priori (Beyer et al., 2024; Zhai et al., 2022; Oquab et al., 2023; Trinh & Le, 2018; Gao et al., 2020; Mizrahi et al., 2025; Allal et al., 2025; OLMo et al., 2024). In practice, however, this is unrealistic, since this assumes prior knowledge of all possible test sets that the model may encounter in the wild. Infact, several popular language model trainers do not decontaminate their training sets precisely for this reason (Su et al., 2024; Weber et al., 2024; Maini et al., 2025; Rae et al., 2021; Penedo et al., 2023; Kandpal et al., 2025). Further, while we acknowledge that our post-hoc contamination analysis can be limiting and would benefit from a more causal treatment such as in works like (Li et al., 2024; Soldaini et al., 2024; Bordt et al., 2024; Jiang et al., 2024), we however note that the downside of such a causal analysis is the significant overhead of re-training our models. Hence, we also note that many works in the literature refrain from a fully causal treatment of contamination (Radford et al., 2019; Brown et al., 2020; Dubey et al., 2024; Achiam et al., 2023).

**Language-only detection.** Our contamination detection only operates on the seed text-datasets that we generate our synthetic datasets from. We have not done any contamination analysis between the spoken question audio in our test sets with the audio in our training sets (we note that prior works in speech-language processing also mainly do contamination analysis at the text-level (Ngo et al., 2025; Tseng et al., 2025)). While this is a reasonable proxy for our synthetic datasets, such a method might not transfer well for decontamination analyses of web-crawled datasets. This is because many of the speech transcriptions of the web-crawled speech might be noisy, incorrect or contain hallucinations induced by the transcription model. Hence, measuring, detecting and quantifying contamination on the audio modality is an important research problem that warrants futher research attention.

**Testing on non-contaminable benchmarks.** While research in optimal ways to do test-set contamination in language models is still nascent, many works take the alternate approach of building benchmarks that are by construction non-contaminated (Ghosh et al., 2024; White et al., 2024; Zeng et al., 2025; Wildman et al., 2025; Jain et al., 2024; Zhang et al., 2024b; Yang et al., 2023; Srivastava et al., 2024). We note that there is a huge gap in such robust evaluations in the speech-language modeling community, and striving for better benchmarks would enable stronger significance in results, while diminishing the impacts of train-test contamination on downstream model performance.

## L.5 CODE FOR IDENTIFYING MATCHES

**Algorithm 1** PyTorch-style code for identifying contaminated samples

```python
def find_contamination_hits():
    """
    Code to get all n-gram contaminated spans in the train set within a window.
    - Finds all n-grams from all evalsets
    - Finds intersection with all n-grams from training set
    """
    # loading eval texts. Question + Answer combined.
    eval_texts: t.List[str] = load_eval_set_texts()
    # loading trainset texts
    train_texts = load_training_set_texts()
    hits = []
    # we consider a window from 6-gram to 13-gram
    for n in range(6, 14):
        # set of all n-grams from all evalsets
        eval_tokens = set()
        for eval_text in eval_texts:
            # tokenizer used is `tiktoken.encoding_for_model("gpt-4o")`
            tokenized = tokenizer(eval_text)
            for i in range(n, len(tokenized)):
                cur_window = tokenized[i-n:i]
                eval_tokens.update(",".join(cur_window))

        for train_text in train_texts:
            tokenized = tokenizer(train_text)
            for i in range(n, len(tokenized)):
                cur_window = tokenized[i-n:i]
                if ",".join(cur_window) in eval_tokens:
                    # record hits from training set
                    hits.append(train_text)
                    break

    return hits
```

# M POST-TRAINING DETAILS

## M.1 POST-TRAINING DATA

Our SFT data consists of the following components:

- *Question-and-Answer Conversations*: We start from about $1.5$ million question-answer conversations in text between users and simulated assistants (more details can be found in Section 4.3.2 in Gunter et al. (2024)). We filter out conversations that are not suitable for spoken dialogue (e.g., conversations involving coding or large chunks of math equations) and rewrite the assistant responses to make them more concise. We then use both `melo-TTS` (Zhao et al., 2023) and `gpt4o-audio` to synthesize text conversations into speech. About 1 millon spoken dialogues are generated in this manner. Further, to improve the robustness to voice variations and background noises on the user side, we mined about $500k$ speech segments whose transcription indicates that it is a question that can be answered with the given context. We then generate the text response and synthesize it in speech. Both mining and response generation are done by querying `gpt-4o`. Speech synthesizing is done via `gpt4o-audio`.

- *TTS and ASR-style Conversations*: We convert utterances from ASR/TTS datasets into natural conversation, in which users ask assistants to either transcribe a given audio (ASR) or synthesize a given text (TTS). We also include instruction-following TTS data where users ask to synthesize text responses with specific instructions (e.g., synthesize speech in a given volume, pace, style or emotion).

- *Conversations with emotion and general audio understanding knowledge*: Here, we include spoken conversations where users ask assistants questions that require emotion, sound and music understanding. As before, we generate such conversations using `gpt-4o` and synthesize speech using `gpt4o-audio`.

## M.2 SFT TRAINING DETAILS

For SFT training, we used a constant learning rate of $5e-5$ with $0.1$ dropout. We train for 20k steps using a batch size of $256$ and sequence length of $16,384$. To prevent regression on text-related metrics, we mix in a text pre-training dataset with a $0.6$ sampling weight, i.e., 40% of the joint SFT mix is audio SFT data.

Unlike pretraining, we found it useful to explicitly generate the chain-of-thought trajectory, i.e., before the model generates assistant's audio response for the $t$-th turn, $A_t^a$, we ask the model to generate text tokens for what the user has said in the $t$-th turn, $T_t^u$, and what assistant would say in text, $T_t^a$. Therefore, for a $T$-turn conversation, $(A_1^u, T_1^u), (A_1^a, T_1^a), \cdots, (A_T^u, T_T^u), (A_T^a, T_T^a)$, we formulate a sequence, $\underline{A_1^u}, T_1^u, T_1^a, A_1^a, \cdots, \underline{A_T^u}, T_T^u, T_T^a, A_T^a$. The loss from users' audio tokens (those marked with underlines) are masked out during training.

## M.3 SFT EVALUATION DETAILS

**Text response quality.** We use two evaluation datasets: *spoken-alpaca* and *noisy-alpaca*. The first is obtained by synthesizing the alpaca evaluation dataset (Li et al. (2023)). On top of *spoken-alpaca*, we added various background noise with a SNR randomly sampled from 5 to 15 dB. This produces *noisy-alpaca*. During the evaluation, 804 spoken alpaca questions were fed in, and the model's text response, $T_1^a$, is extracted. These text responses are pair-wise compared with the responses generated from a performant internal baseline model using the standard evaluation protocol with `gpt-4o-mini-2024-07-18` as the judge model.

**Audio response quality.** To evaluate audio response quality, we work with several third-party vendors to collect diversified user prompts in audio. For multi-turn dialogue evaluation, we adopt the last-turn-with-context strategy to evaluate the last turn's assistant response, while the previous assistant responses are generated by `gpt-4o-audio` and fed in as context. In total, we constructed 5 evaluation sets, each having a different focus, such as knowledge-rich, multi-turn, long-context, and challenging speech environments. We also notice pair-wise comparison of audio is often harder than text, in which judges (LLM or human) cannot tell which response is better. In order to reduce

variance of judge scores, we ask the judge to output whether audio response A is better than, worse than or tied with audio response B. The auto-grading prompt template we used is in Appx. I.

# N  PRELIMINARY EVALUATIONS ON SPEECH-TO-SPEECH TASKS

We emphasize that the primary goal of our work is to conduct a clean, controlled empirical study of data-centric choices for improving spoken question-answering in the speech-to-text (S→T) setting. We focus on S→T for three reasons: (i) S→T benchmarks have more mature and structured evaluation protocols, which makes them well-suited for targeted ablations, (ii) producing correct text outputs is a necessary first step before meaningfully assessing speech-to-speech (S→S) generation, and (iii) S→S evaluation is considerably more delicate, as it can entangle semantics (*what is said*) with acoustics (*how it is said*). In particular, using ASR to convert S→S outputs back to text introduces additional error from the ASR system, whereas directly relying on speech-token log-likelihoods risks over-emphasizing acoustic fidelity relative to semantic correctness.

We however believe S→S evaluation is also an important dimension to measure. As a first step, we hence evaluated our models on Sblimp and StoryCloze (both Spoken [SSC] and Topic [TSC] variants). The results for each individual setting studied in Sec. 3 are presented below.

| Interleaving Granularity | Sblimp | SSC | TSC |
|---|---|---|---|
| Coarse | 54.1 | 51.3 | 73.2 |
| Fine | 54.5 | 53.3 | 73.5 |

| Data Mix | Sblimp | SSC S→S | TSC S→S |
|---|---|---|---|
| Web-crawl 100% | 54.5 | 53.3 | 73.5 |
| Web-crawl 53% + Krist 47% | 54.6 | 52.7 | 74.1 |
| Web-crawl 66% + Quest 34% | 54.4 | 51.6 | 73.0 |
| Web-crawl 59% + Quest 6% + Krist 35% | 54.5 | 52.0 | 73.3 |
| Web-crawl 40% + Quest 27% + Krist 33% | 54.4 | 51.8 | 73.2 |

| Sampling Scheme | Sblimp | SSC S→S | TSC S→S |
|---|---|---|---|
| Stochastic | 54.4 | 51.8 | 73.2 |
| Deterministic | 54.5 | 51.7 | 69.1 |

We mainly find:

1. Fine interleaving outperforms coarse interleaving on both Sblimp and Storycloze evaluations. This emphasizes that our main takeaways regarding the interleaving strategy holds true even for speech-to-speech evaluations.

2. For the synthetic data variants, the results are not fully conclusive. We note that the Web-crawl 53% + Krist 47% checkpoint improves on TSC while degrading on SSC. For the other checkpoints, on SSC most of them seem to drop, whereas for TSC the drop seems to be much more minor.

3. For the deterministic vs stochastic experiment, we note that both checkpoints used synthetic data in the training mix, and due to this it is hard to draw conclusions as to whether there is a large improvement in favour of either strategy.

Finally, we note that our synthetic data uses different voices from the natural evaluation data, which can affect audio-token log-likelihoods and thus these S→S scores. This is precisely the challenge we mentioned above regarding the coupling of semantics and acoustics in S→S evaluation. Nonetheless, our preliminary results suggest that synthetic data does not collapse the model's speech generation capabilities, at most, it leads to mild degradations.

## O COMPARISON TO CASCADED BASELINE

In this section, we also include comparisons to a two-stage pipeline that first produces the text transcriptions of the test set audio question using Whisper-v3-large, and then decodes the final answer using the *Text-init* language model (i.e. the original language model we start continued pretraining from). We observe that the simple cascaded pipeline outperforms all the SpeechLMs we compared in the main paper. This observation is consistent with recent results in the SpeechLM literature (Sakshi et al., 2024; Cui et al., 2025) showcasing that speech-language cascades (that first transcribe the question from audio and then process the text using a language model) outperform end-to-end speech-language models. However, our `SpeLangy` model does close the gap to the performance of the cascade, especially in the Spoken-Web-Questions evaluation, highlighting that we are making progress towards closing the gap between end-to-end speech-language models and cascaded systems.

Table 16: **Spoken Question-Answering (S→T) comparison with cascade baseline.**

| Type | Model | # Params | SWQ | STQ | SLQ | Average |
|---|---|---|---|---|---|---|
| **Cascade** | Whisper-v3-large + Text-init | – | 51.1 | 68.8 | 76.3 | 65.4 |
| **Base** | Kimi-Audio | 10.5B | 44.0 | 33.8 | 47.0 | 41.6 |
| | Qwen-Audio | 8.4B | **45.7** | 30.3 | 46.0 | 40.7 |
| | Qwen-2-Audio | 8.4B | **45.7** | 33.4 | 47.0 | 42.0 |
| | `SpeLangy` | 3.8B | **45.7** | **44.6** | **65.0** | **51.8** |
| **SFT** | Voxtral-mini | 4.7B | 41.6 | 46.6 | 65.3 | 51.2 |
| | GLM-4-Voice | 9.9B | 43.3 | 52.4 | 64.7 | 53.4 |

## P    LIMITATIONS AND FUTURE DIRECTIONS

While we conducted extensive experiments to study the three data-centric questions outlined in Fig. 1, there are still a few limitations in our work that can be improved upon:

**Model sizes and compute budgets.**    All our experiments were at the 3.8B parameter scale trained for 1.67T speech-text tokens (roughly $\sim 3.81 \times 10^{22}$ FLOPs). While our results are strong (outperforming models that are $3\times$ the size, trained for similar compute budgets), it would still be interesting to explore if our data-centric strategies would hold at larger model scales. While recent papers like Nezhurina et al. (2025), DataComp-LM (Li et al., 2024), HoneyBee (Bansal et al., 2025) and DataComp-CLIP (Gadre et al., 2023) suggest transferability of data curation methods across model scales, recent work in language and vision-language modeling has posited that there may be trade-offs when applying data curation across different model sizes and compute budgets (Mizrahi et al., 2025; Goyal et al., 2024). To the best of our knowledge, no existing work showcases such trade-offs in the SpeechLM community. It would be an interesting direction to explore the interaction of data recipes with model scale and compute budget.

**More speech-text tasks.**    Since the focus of our work was mainly on improving spoken question-answering capabilities of SpeechLMs, all our experiments used the standard benchmarks that are prevalent in the literature for our task of interest (Liu et al., 2025; Xiaomi, 2025; Li et al., 2025c; Ding et al., 2025). We therefore did not explore how our models would perform on more targeted tasks like automatic speech recognition, emotion recognition or text-to-speech synthesis. One caveat preventing us from a direct comparison on such tasks is that we do not employ any task-specific training, unlike other SpeechLMs that explicitly add in a task-specific component into their training mixture (e.g., ASR-specific training datasets) (Li et al., 2025c; Ding et al., 2025; Liu et al., 2025).

**End-to-end evaluation.**    Currently, our evaluations involve testing on text-only benchmarks (text-in text-out) and spoken question-answering benchmarks (audio-in text-out). However, end-to-end spoken question-answering, where both the input and output is in audio (audio-in audio-out) is an important capability that remains untested. While there have been some prior works testing explicitly for the full end-to-end capability (Ding et al., 2025; Li et al., 2025c; Hassid et al., 2023; Xiaomi, 2025; Nachmani et al., 2023), we note that reliable evaluation for this task is still quite challenging—there is a lack of standardization in the evaluation procedures used across the different model releases. For example Kimi-Audio (Ding et al., 2025) uses a human judgement rating for comparing model outputs, while GLM-4-Voice (Zeng et al., 2024a), MiMo-Audio (Xiaomi, 2025), Spectron-LM (Nachmani et al., 2023) and Baichuan-Audio (Li et al., 2025c) use automated methods with ASR transcription models and LLM-as-judges. However, the ASR and judge-models used can be biased and impact results quite a lot (Ye et al., 2024b; Panickssery et al., 2024), which has not been discussed in these prior works. More importantly, previous works in image omni-models have demonstrated that the data curation procedures for targeting understanding and generation capabilities might differ significantly (Tong et al., 2024b; Chen et al., 2025; Deng et al., 2025; Wu et al., 2025b; Zhang et al., 2025). Hence, we posit that similar takeaways might also hold for the speech-language pretraining task, where the data processing and curation strategies for understanding only tasks (audio-in text-out) are potentially different from generation tasks (audio-in audio-out). However, it is an interesting and important direction to test if our approaches transfer to the full end-to-end evaluation setting as well.

**Few-shot capabilities.**    Currently, all our evaluations for spoken question-answering used a 0-shot prompting strategy i.e. the model would be fed in an input audio question and has to respond in text, with no additional examples in-context. However, many of the text-only evaluations including MMLU and WebQuestions are few-shot / in-context evaluations (MMLU is 5-shot and WebQuestions is 1-shot). Evaluating our models' abilities in the few-shot / in-context setting can further yield important insights on transferability and steerability of our models. Importantly, the few-shot capability has been emphasized to large degrees in both the vision-language (Zhou et al., 2022; Zhang et al., 2021b; Gao et al., 2024b; Udandarao et al., 2023; Alayrac et al., 2022; Awadalla et al., 2023; Laurençon et al., 2024) and text-only (Brown et al., 2020; Achiam et al., 2023; Touvron et al., 2023; Dong et al., 2022; Olsson et al., 2022) foundation modeling literature. Recently, MiMo-Audio (Xiaomi, 2025) also described their experimental settings which included few-shot speech-text tasks. Studying the transfer of our data interventions to the few-shot evaluation setting is an important open problem.

**Training from scratch.** All our training runs initialize the language model backbone for our SpeechLM using a pretrained base-LM. This is the standard recipe used by almost all the existing foundation SpeechLMs (Li et al., 2025c; Défossez et al., 2024; Liu et al., 2025; Wu et al., 2025a; Xiaomi, 2025; Chu et al., 2024; Ding et al., 2025; Zeng et al., 2024a). However, recent work in the vision-language literature has advocated for full native multimodal pretraining from scratch (Shukor et al., 2025), where both the language model and the modality-specific encoder/tokenizer are trained from scratch. It would be interesting to explore if our data-centric methods also enable more efficient SpeechLM pretraining from scratch in the future.

**Better training recipes.** In all our experiments, we freeze the speech tokenizer while only training the language model. In the SpeechLM literature, there is no strong consensus regarding freezing or unfreezing the speech tokenizer. A potential next step could be to unfreeze the tokenizer and study the transferability of our data-centric recipes. Additionally, we conduct only one continued-pretraining stage—however, recent SpeechLM works have explored more sophisticated multi-stage pipelines involving pretraining and mid-training (Wu et al., 2025a; Xiaomi, 2025; Li et al., 2025c; Goel et al., 2025). It would again be interesting to test our methods in a multi-stage pipeline.

**Better data mixtures.** In our experiments, we always used a mixture ratio of 60% text and 40% speech-text tokens. While we followed existing multimodal literature for these ratios (Shukor et al., 2025; McKinzie et al., 2024; Tong et al., 2024a), it is likely that this mixture ratio could be further tuned. A key reason for having such a large text-only proportion was to ensure the model does not lose its language-only base capabilities. However, for larger models (7B-parameter scales and beyond), a smaller text-proportion might be viable since larger models generally are prone to lesser catastrophic forgetting (Yıldız et al., 2024; Roth et al., 2024; Dziadzio et al., 2025; Ramasesh et al., 2021; Ibrahim et al., 2024). Indeed, recent SpeechLMs like MiMo-Audio (Xiaomi, 2025) and StepAudio-AQAA (Huang et al., 2025) use much smaller text-proportions in their training mix, suggesting that this is a valid strategy to improve speech-language pretraining.

