# OpenReview forum: "Data-Centric Lessons To Improve Speech-Language Pretraining"
_ICLR.cc/2026/Conference — ICLR 2026 Poster_

### Official Review · Reviewer_jiGx · 2025-10-30

**Soundness:** 3
**Presentation:** 3
**Contribution:** 2
**Rating:** 4
**Confidence:** 4

**Summary:**

This paper presents a systematic data-centric study on improving speech-language pretraining for spoken question answering (SQA). The authors focus on how data processing and curation affect performance, exploring three key questions: how to process raw web audio, how to construct synthetic speech-text data, and how to interleave modalities during training. Through controlled experiments, they show that fine-grained interleaving, synthetic datasets generated via LLM rewriting and text-to-speech synthesis, and deterministic speech-text sampling each yield performance gains. Integrating these insights, their 3.8B-parameter model SpeLangy surpasses models in SQA accuracy, underscoring that careful data design is crucial for advancing SpeechLMs.

**Strengths:**

1. The paper raises a critical point about pretraining Speech-Language Models (SLMs): although prior works mention the importance of data quality, few have conducted controlled ablations to determine which data factors drive performance. This study fills that gap with systematic and well-designed analyses.
2. The paper is clearly structured and easy to follow. Each data-centric question is introduced with clear motivation, supported by quantitative results that highlight the impact of each design choice.
3. Building on the insights from the proposed data interventions, the authors present **SpeLangy**, a 3.8B SpeechLM that consistently outperforms much larger baselines. This demonstrates the practical significance and scalability of their data-centric approach.

**Weaknesses:**

1. While SQA is a suitable benchmark for evaluating multimodal alignment, the scope of evaluation could be expanded. Including open-ended spoken reasoning or dialogue tasks would better demonstrate the generalizability of the proposed data curation strategies beyond structured QA.
2. The empirical benefits of some interventions are relatively modest. For example, as shown in Table 2, gains from incorporating *Krist* are minor, and the difference between stochastic and deterministic modality sampling (Table 3) is small. A deeper analysis of why certain methods contribute less or under which conditions they are more effective would strengthen the overall conclusions.

**Questions:**

1. Several supporting materials, such as **Figure 8** and **Table 11**, are placed far from the sections where they are first referenced. Could the authors consider moving or cross-referencing them more clearly to improve readability and flow?
2. Have the authors examined whether the proposed data-centric methods (e.g., fine-grained interleaving, deterministic sampling) interact synergistically, or whether their gains are largely independent? Clarifying this would help readers understand which factors are most crucial for reproducing the improvements.

**Details Of Ethics Concerns:**

The paper leverages large-scale web-crawled data for model pretraining but does not specify how data collection complies with copyright, licensing, or web-crawling policies. Clarification on data provenance, consent, and compliance with legal and ethical standards (e.g., GDPR) is needed.

---

> ### Author Response · Authors · 2025-11-22
> **Rebuttal**
>
> We thank the reviewer for recognizing and highlighting the importance of our controlled data-centric analysis and for noting the clarity of the paper. We address your individual concerns point-by-point below.
>
> > **W1. Scope of evaluation can be expanded.**
>
> We thank the reviewer for this suggestion. We mainly focused on SQA for three reasons:
> 1. It is a *core real-world use case* (voice assistants, spoken search, tutoring, etc.).
> 2. It provides a *mature, standardized evaluation protocol* with multiple benchmarks and strong baselines.
> 3. It allows a *clean experimental setup*: we use only interleaved speech–text continued pretraining, avoiding additional ASR/S2T/TTS losses that could confound the effect of data curation (task interference, data-mix ratios, etc.).
>
> However, we agree with the reviewer that expanding the evaluation scope to further open-ended dialogue and chat tasks is important. To address this concern, we show that indeed optimizing for SQA during pretraining also transfers to real-world chat evaluations like Alpaca-Eval. We conduct a very lightweight SFT stage from our pretrained checkpoints (details in the updated draft in section 3.7) on mixed speech–text conversational data and evaluate responses using LLM-as-judge metrics for *both text and audio quality* (using win-rates, details in appendix L of the updated draft). We use three pretrained checkpoints:
>
> - *Coarse*: coarse interleaved baseline only trained on web-crawled data (row 1 of table 1 in draft).
> - *Fine*: fine interleaved model only trained on web-crawed data (row 2 of table 1 in draft).
> - *Fine + Syn*: fine interleaved model trained with Quest data (best model in table 2 of the draft).
>
> We provide results in the table below (and have added them as table 5 in the revised draft).
>
> | Pretrain ckpt |      Text quality       |               |         Audio Quality          |        |        |        |        |
> |--------------:|------------------------:|--------------:|:------------------------------:|:------:|:------:|:------:|:------:|
> |               | spoken-alpaca          | noisy-alpaca  | Eval 1                         | Eval 2 | Eval 3 | Eval 4 | Eval 5 |
> | coarse        | 42.6                   | 45.2          | 37.4 (17.2)                    | 33.3 (24.1) | 34.3 (18.1) | 37.0 (16.3) | 38.8 (16.9) |
> | fine          | 44.3                   | 47.3          | 39.9 (18.5)                    | 33.8 (23.7) | 36.4 (11.6) | 38.0 (16.9) | **41.9 (20.7)** |
> | fine + syn    | **47.4**               | **48.8**      | **41.1 (17.1)**                | **36.6 (23.1)** | **40.1 (18.7)** | **39.4 (16.9)** | 39.3 (16.8) |
>
> We indeed find that the models trained with **our best data configurations (fine interleaving and synthetic data) are preferred in the majority of pairwise comparisons** and **never underperform the baseline configuration**, indicating that higher SQA accuracy correlates with better downstream conversational quality after SFT.
>
> We have added these new results in the revised draft showcasing that our data choices indeed **generalize to real-world speech tasks**. We thank the reviewer once again for this suggestion, and believe it has significantly extended the generality and significance  of our findings.

---

> ### Author Response · Authors · 2025-11-22
> **Rebuttal contd**
>
> > **W2. Modest gains from some methods, deeper analysis?**
>
> We agree that not all interventions yield large gains, and we see this as a realistic outcome of *data-centric work as an empirical science* rather than a weakness. However, we provide some hypotheses in our work about why certain methods might be yielding gains.
>
> 1. *Question extraction and formatting as an implicit quality filter.* Quest employs a question extraction step unlike Krist which might implicitly act as a *quality filter*. Further, the formatting difference between Krist and Qust might also lead to different induced performance differences. We think it is an interesting direction in the future to decouple quality and format along separate axes of study, similar to prior recent works in language modeling ([Wettig et al, 2025](https://arxiv.org/abs/2502.10341)).
>
> 2. *Granularity of fine-grainedness.*  Both fine interleaving and deterministic sampling effectively *reduce average audio-chunk length*, giving the model more, shorter cross-modal alignments; however, the length shift is **much larger for fine vs. coarse interleaving than for deterministic vs. stochastic sampling**, which could potentially explain the difference in their impact.
>
> We believe our KL-divergence analysis can further help shed light on these hypotheses. Methods that reduce the divergence between text-conditioned and speech-conditioned output distributions might help the most. We therefore suggest that tracking the KL-divergence metric in the future might be beneficial to better understand the efficiency of a particular data-intervention. Indeed, since the ICLR deadline, we have found that one paper ([Cuervo et al, 2025](https://arxiv.org/abs/2510.13632)) adopts a similar KL-divergence metric to quantify the gap between speechLM and base-LM downstream performance, that is directly correlated to the end SpeechLM quality. Hence, we believe that our main results along with the conducted analyses **offer strong empirical insights that can be adopted by SpeechLM practitioners**.
>
> > **Q1. Better placement and cross-referencing fo figures and tables.**
>
> We apologize for this oversight. In the revised version of the draft, we have moved the contamination table to the section where it is referenced (in the updated draft, it is Figure 7). Further, due to lack of space, we did not move the data processing pipeline figure to the main paper, but cross-reference it better in the text, highlighting that it is in the appendix (please see the blue text in the corresponding section in the revised draft).
>
> > **Q2.  Interactions between data-centric methods.**
>
> We thank the reviewer for this important question. While we agree that a more systematic analysis of interactions between the different data-methods would help provide a more nuanced understanding, conducting such an analysis, in our eyes, would be computationally very expensive and impractical. In our experiments, we followed a "stack-the-best" philosophy – for every experiment, we built on top of the previous best settings. First, we identify whether fine or coarse-grained interleaving yields benefits, having established that, we took the best setting—fine-interleaving—and used that for all subsequent experiments. This is also the typical approach taken in established data-centric studies like ([Guha et al, 2025](https://arxiv.org/abs/2506.04178), [Li et al, 2024](https://arxiv.org/abs/2406.11794), [Gadre et al, 2023](https://arxiv.org/abs/2304.14108)), and hence we believe it is a justified approach. However, we agree with the reviewer that understanding such interactions would be a great avenue for future research.

---

> > ### Author Response · Authors · 2025-11-22
> > **Rebuttal contd**
> >
> > > **Ethics concern.**
> >
> > We appreciate this important point and have added a dedicated ethics section with the following clarifications:
> >
> > - **Data provenance.** All speech data comes from **publicly available podcast RSS feeds and similar spoken-word streams**. We do not scrape behind paywalls and avoid clearly copyrighted catalogue content such as commercial audiobooks and music albums.
> > - **Web-crawling policies.** Our collection framework respects **`robots.txt` directives** and website-specific terms of use.
> > - **Licensing.** We preferentially include sources under **permissive or podcast-typical licenses** that allow redistribution for research. When license information is ambiguous, we err on the side of exclusion.
> > - **Privacy & PII.** We apply automatic filters to reduce personally identifiable information (e.g., email addresses, phone numbers) and run safety classifiers to remove clearly harmful or sensitive content. We have used this data for research purposes under our institution’s data processing policies.
> > - **GDPR and regional compliance.** Processing is conducted in accordance with internal legal guidance, and only aggregate, non-identifiable statistics are reported. No attempt is made to profile or target individual speakers.
> >
> > We will make these points explicit in the ethics section and data appendix.
> >
> > **We once again thank the reviewer for their efforts in carefully evaluating our paper, it is greatly appreciated! We believe your suggestions have greatly improved the quality of our paper, and hope that our responses have sufficiently addressed your concerns. We look forward to your response and further discussion!**

---

### Official Review · Reviewer_QDUW · 2025-10-30

**Soundness:** 3
**Presentation:** 3
**Contribution:** 2
**Rating:** 4
**Confidence:** 3

**Summary:**

This work explores three key issues regarding data construction and usage in building large speech models. The authors find that: 1) Fine-grained mixing of speech and text tokens helps improve cross-modal learning; 2) Training speech LLMs with synthesized data collected from websites is beneficial; 3) Deterministic sampling for mixing speech and text data yields better results than random sampling. Based on these findings, the authors trained a 3.8B parameter model and evaluated it on a spoken question answering task, achieving performance comparable to other larger open-source models.

**Strengths:**

The article is clearly structured and easy to follow. It provides an analysis of the construction details of speech data in large speech models, a topic that has not been extensively explored in existing works.

**Weaknesses:**

1. Overall, this work remains largely analytical, and the answers to several questions are relatively straightforward. For instance, the conclusion that "fine-grained interleaved tokens are better than coarse-grained" is not particularly surprising, as mainstream approaches in speech-to-speech LLMs already employ word-level interleaved strategies, which are inherently fine-grained. Therefore, this finding does not significantly impact the development of large speech models, and its contribution is limited. This paper might be more suitable for workshops or conferences focused on data engineering.

2. There is a lack of comparison between the proposed foundation model and other text-based models to determine whether its performance is competitive. Furthermore, the evaluation is limited to a spoken question-answering task, which is not comprehensive enough to be fully convincing.

3. The experimental comparisons are not entirely fair. The proposed "SpeLang" model is designed for speech-to-text tasks, while the compared models (e.g., GLM-4-Voice, Kimi Audio) are primarily for speech-to-speech. For Qwen2 Audio, the model is designed for a broader range of inputs (speech, sound, and music). Therefore, the comparison is not conducted on a level playing field.

**Questions:**

1. Since you are investigating a speech-to-text model, what is its performance on traditional tasks such as Automatic Speech Recognition (e.g., on LibriSpeech) and Speech Translation (e.g., on CoVoST)? How does this performance compare to other specialized models?

2. In Line 292, what is the "audio-loss-masked" setting? This concept lacks a detailed description in the paper.

---

> ### Author Response · Authors · 2025-11-22
> **Rebuttal**
>
> We appreciate the reviewer’s comments on clarity and the recognition that our work addresses a key underexplored aspect of SpeechLMs. We respond to your concerns about novelty, evaluation scope, and fairness below.
>
> > **W1. Findings are largely analytical and not surprising, limited contribution.**
>
> We fully agree that some of our *qualitative* conclusions might align with community intuition. However, we want to emphasize that our main contributions are to:
>
> 1. **Revisit and thoroughly test data decisions at web scale under modern setups.**
> We agree with the reviewer that earlier models like [SpiritLM](https://arxiv.org/abs/2402.05755) used **word-level interleaving on relatively small, clean, transcribed datasets**. In contrast, we note that current SpeechLMs train on **multi-million-hour web data** with noisy ASR transcriptions and diverse content. Further, these recent models such as [Kimi-Audio](https://arxiv.org/abs/2504.18425), [Baichuan-Audio](https://arxiv.org/abs/2502.17239), and [GLM-4-Voice](https://arxiv.org/abs/2412.02612) use **coarser interleaving strategies at the sentence-level** that **do not operate at the word-level**. Hence, this leaves open whether fine interleaving remains optimal under these new conditions.
>
> 2. **Provide controlled, apples-to-apples comparisons.**
> We design a **single clean pretraining pipeline** and vary **only one factor at a time** (interleaving granularity, synthetic mix, sampling scheme) while holding everything else fixed. This allows us to **quantify the impact** of each decision, rather than infer it from incomparable model releases. **To the best of our knowledge, we are the first to systematically compare different data strategies for speech-language interleaving strategies, on a level-playing field.**
>
> 3. **Offer analysis tools, not just empirical scores.**
> Beyong our controlled empirical comparisons, we also provide some insights into *why* certain methods might yield improved performance. For instance, we provide:
>    - KL divergence analysis between text-conditioned and speech-conditioned distributions of the same LM,
>    - statistics of audio chunk length distributions,
>    - topic coverage and contamination analyses.
>
>    These reveal, for example, that fine interleaving reduces the KL gap between speech and text conditioning, and that deterministic sampling can increase the number of modality switches seen during training. **Such an analysis does not only explain some of our results, but also provides deeper insights for future data-centric approaches.**
>
> We therefore view our main contribution as **practical, data-centric guidance** backed by quantitative analysis and large-scale experiments, consistent with recent data-centric works at ICLR like [Hu et al, 2023](https://arxiv.org/abs/2309.16671), [Elazar et al, 2023](https://arxiv.org/abs/2310.20707), [Mayilvahanan et al, 2025](https://arxiv.org/abs/2410.08258) and [Fang et al, 2023](https://arxiv.org/abs/2309.17425). We have updated the text in the last paragraph of the introduction to precisely highlight the scope and contribution of our work (in blue text).
>
> > **W2. Lack of comparison vs. text-only LMs**
>
> We respectfully point out that we already compare SpeLangy against strong text-only LMs like Gemma-2, Gemma-3 and Qwen-2.5 in Table 7 of the paper. We show that SpeLangy **remains competitive** despite continued pretraining on speech–text data. We sincerely apologize if this was missed in our previous draft due to an oversight on our part. We have revised the caption of Table 7 and surrounding text to highlight more clearly that this table is comparing SpeLangy with **strong text-only LMs** (the edits to the caption and text are highlighted in blue).

---

> ### Author Response · Authors · 2025-11-22
> **Rebuttal contd**
>
> > **W2. Evaluation is limited to SQA, generality of findings?**
>
> This is a great question. We choose SQA as our main evaluation testbed due to three primary reasons:
> - It is a *core real-world use case* (voice assistants, spoken search, tutoring, etc.).
> - It provides a *mature, standardized evaluation protocol* with multiple benchmarks and strong baselines.
> - It allows a *clean experimental setup*: we use only interleaved speech–text continued pretraining, avoiding additional ASR/S2T/TTS losses that could confound the effect of data curation (task interference, data-mix ratios, etc.).
>
> To address the concern about generality and comprehensivenss of evals, we show that indeed optimizing for SQA during pretraining also transfers to real-world chat evaluations like Alpaca-Eval. We conduct a very lightweight SFT stage from our pretrained checkpoints (details in the updated draft in section 3.7) on mixed speech–text conversational data and evaluate responses using LLM-as-judge metrics for *both text and audio quality* (using win-rates, details in appendix L of the updated draft). We use three pretrained checkpoints:
>
> - *Coarse*: coarse interleaved baseline only trained on web-crawled data (row 1 of table 1 in draft).
> - *Fine*: fine interleaved model only trained on web-crawed data (row 2 of table 1 in draft).
> - *Fine + Syn*: fine interleaved model trained with Quest data (best model in table 2 of the draft).
>
> We provide results in the table below (and have added them as table 5 in the revised draft).
>
> | Pretrain ckpt |      Text quality       |               |         Audio Quality          |        |        |        |        |
> |--------------:|------------------------:|--------------:|:------------------------------:|:------:|:------:|:------:|:------:|
> |               | spoken-alpaca          | noisy-alpaca  | Eval 1                         | Eval 2 | Eval 3 | Eval 4 | Eval 5 |
> | coarse        | 42.6                   | 45.2          | 37.4 (17.2)                    | 33.3 (24.1) | 34.3 (18.1) | 37.0 (16.3) | 38.8 (16.9) |
> | fine          | 44.3                   | 47.3          | 39.9 (18.5)                    | 33.8 (23.7) | 36.4 (11.6) | 38.0 (16.9) | **41.9 (20.7)** |
> | fine + syn    | **47.4**               | **48.8**      | **41.1 (17.1)**                | **36.6 (23.1)** | **40.1 (18.7)** | **39.4 (16.9)** | 39.3 (16.8) |
>
> We indeed find that the models trained with **our best data configurations (fine interleaving and synthetic data) are preferred in the majority of pairwise comparisons** and **never underperform the baseline configuration**, indicating that higher SQA accuracy correlates with better downstream conversational quality after SFT.
>
> We have added these new results in the revised draft showcasing that our data choices indeed **generalize to real-world speech tasks**. We hope that these new results showcase the generality of our data approaches across different important tasks of interest.

---

> ### Author Response · Authors · 2025-11-22
> **Rebuttal contd**
>
> > **W3. Fairness of comparisons.**
>
> We thank the reviewer for this question. We first want to highlight that the models we compare against are the only SoTA openly-released models as of the time of release, and hence comparing against them enables us to demonstrate the effectiveness of our data-centric findings.
>
> To further address the concerns of fairness of comparisons, we show two more results:
>
> 1. **SpeLangy remains competitive when compared to SFT variants.**
> We compare SpeLangy against SFT-tuned variants of the baselines (e.g., Kimi-Audio-Chat, Qwen-Audio-Instruct, Qwen2-Audio-Instruct) and find that SpeLangy matches or is competitive with these SFT models on SQA. This showcases that our data recipes help improve SQA performance beyond simply instruction tuning and QA-like formatting.
>
> | Type | Model         | #Ps  | SWQ  | STQ  | SLQ  | Average |
> |------|---------------|------|------|------|------|---------|
> | Base | Kimi-Audio    | 10.5B| 44.0 | 33.8 | 47.0 | 41.6    |
> | Base | Qwen-Audio    | 8.4B | 45.7 | 30.3 | 46.0 | 40.7    |
> | Base | Qwen-2-Audio  | 8.4B | 45.7 | 33.4 | 47.0 | 42.0    |
> | Base | SpeLangy      | 3.8B | 45.7 | 44.6 | 65.0 | 51.8    |
> | SFT  | Voxtral-Mini  | 4.7B | 41.6 | 46.6 | 65.3 | 51.2    |
> | SFT  | GLM-4-Voice   | 9.9B | 43.3 | 52.4 | 64.7 | 53.4    |
> | SFT  | Kimi-Audio    | 10.5B| 47.5 | 40.4 | 51.3 | 46.4    |
> | SFT  | Qwen-Audio    | 8.4B | 50.7 | 33.4 | 67.0 | 50.3    |
> | SFT  | Qwen-2-Audio  | 8.4B | 50.7 | 40.5 | 62.3 | 51.2    |
>
> 2. **SpeLangy when designed for speech-to-speech remains competitive.**
> As the reviewer rightly pointed out, our SpeLangy model is mainly designed for speech-to-text tasks due to the audio loss-masking. However, we show that even when designed for speech-to-speech tasks i.e. without loss masking on audio tokens during interleaved training (called SpeLangy-S2S), the performance of SpeLangy-S2S, while achieving lower performance than the original variant of SpeLangy, still remains competitive with the other baseline models (47.9% on average, outperforming Kimi-Audio, Qwen-Audio and Qwen-2-Audio). In the table below, we highlight for each model, whether they support S2T mode only or also S2S. This result further demonstrates the generality of our data-centric findings, applying to both speech-to-text-only models as well as extending to the speech-to-speech case.
>
> | Type | S2T | S2S | Model         | #Ps  | SWQ  | STQ  | SLQ  | Average |
> |------|------|------|---------------|------|------|------|------|---------|
> | Base | ✔ | ✔ | Kimi-Audio    | 10.5B| 44.0 | 33.8 | 47.0 | 41.6    |
> | Base | ✔ | X | Qwen-Audio    | 8.4B | 45.7 | 30.3 | 46.0 | 40.7    |
> | Base | ✔ | X | Qwen-2-Audio  | 8.4B | 45.7 | 33.4 | 47.0 | 42.0    |
> | Base | ✔ | X | SpeLangy      | 3.8B | 45.7 | 44.6 | 65.0 | 51.8    |
> | Base | ✔ | ✔ | SpeLangy-S2S      | 3.8B | 42.7 | 34.7 | 66.3 | 47.9    |
> | SFT  | ✔ | X | Voxtral-Mini  | 4.7B | 41.6 | 46.6 | 65.3 | 51.2    |
> | SFT  |  ✔ | ✔| GLM-4-Voice   | 9.9B | 43.3 | 52.4 | 64.7 | 53.4    |
>
> > **Q1. Performance on traditional tasks like ASR?**
>
> Our pretraining pipeline **explicitly excludes ASR/S2T-style supervised data and losses**: we purely use speech-text interleaving. Since our goal is to provide a clean empirical setup to study data-centric recipes for SQA, we believe adding such additional tasks like ASR/S2T into the training mixture will lead to confounds due to challenges with task-interference and data-mixing ratios.
> As a result, we do **not** expect SpeLangy to match specialized ASR/ST systems on LibriSpeech or CoVoST, and we think this is **out of scope for the current work**. We have added more text at the end of our introduction section (in blue) to highlight this focus. However, we agree with the reviewer that future work should investigate including different tasks during the speech-language pretraining phase, and how that interacts with data recipes. We have therefore added this as an explicit goal that future work should target in Appendix M of our paper (see new text in blue).
>
> > **Q2. Clarification regarding audio-loss-masked setting.**
>
> We apologize for the lack of clarity in the draft. The audio-loss-masked setting simply means that we apply loss masking on the audio tokens while doing speech-text interleaved training. This is a common setting applied across several recent papers (Kimi-Audio, Baichuan-Audio, Qwen2-Audio, Voxtral). We have modified the text in section 3.6 to make this definition more explicit (please see newly modified text in blue).
>
> **We once again thank the reviewer for their efforts in carefully evaluating our paper, it is greatly appreciated! We believe your suggestions have greatly improved the quality of our paper, and hope that our responses have sufficiently addressed your concerns. We look forward to your response and further discussion!**

---

### Official Review · Reviewer_Yb68 · 2025-10-30

**Soundness:** 2
**Presentation:** 3
**Contribution:** 3
**Rating:** 4
**Confidence:** 4

**Summary:**

This paper conducts comprehensive empirical study in order to find optimal solutions for 3 design choices in speech-LLM pretraining:
1. how to process web-crawled audio content for speech-text pretraining
2. how to construct synthetic datasets to augment web-crawled data
3. how to interleave speech and text modality for pretraining.
The optimization goal is to improve the performance on spoken QA(SQA) task while remaining the the similar text understanding capability after speech-text pretraining. From the empirical study the authors induct the answers for all three questions and deliver a scalable and effective speech-text pretraining recipe. With these insights, they pretrain a 3.8B-parameter SpeechLM, SpeLangy, that outperforms the larger model on SQA task.

**Strengths:**

Quality:
1. The paper presents a well-controlled ablation study, 1 example is the study on different granularity of chunking and interleaving, deterministic vs stochastic sampling schemes.
2. The paper presents data-driven diagnostics and analysis, for example: modality alignment analysis by KL divergence; topic-coverage analysis; contamination checks with n-gram matching. Such analysis provides deep insights that go beyond the benchmark comparison.

Clarity:
The paper employs proper diagram to explain the steps of data pipeline, offers clear schematics for analysis, highlights key conclusions of each ablation study. The paper is clear and easy to follow.

Significance:
The ablation study on data processing and training schema brings sufficient value provides data-driven and practical insights on how to conduct effective speech-text pre-training.

**Weaknesses:**

1. Task scope. This paper limits the evaluation target in Spoken QA (plus text understanding) while positioning the goal of the proposed method as optimizing speech-text pretraining. There could be doubt whether solely SQA is representative. The author need to somewhat prove that correlation between SQA performance and speech-text pretraining quality.
Author discusses about this in Addendum K:
> One caveat preventing us from a direct comparison on such tasks is that we do not employ any task-specific training, unlike other SpeechLMs that explicitly add in a task-specific component into their training mixture (e.g., ASR-specific training datasets).

    However, the fact that the author evaluates on SQA task contradicts their own point.

2. There's a lack of some significant details that would affect the reproducibility, please find more in Questions section.
3. Baselines are missing from some of the ablation study, please find more in Questions section.

**Questions:**

1. Please provide more details about the backbone LM, has it gone through post-training (SFT, RLHF, etc) or just pre-training? Does the LM support speech tokens natively? Or do you have to extend the vocabulary to support speech tokens?
2. Please provide more details about the speech tokenizer. Does it support multi-lingual input?
3. Please provide baseline in Table 2: the base LM performance on text understanding without any training.
4. Isn't training on Quest dataset effectively SQA task-specific finetuning, when the training sample is organized and constructed in the same format as SQA task? Is it fair to compare SpeLangy with other "base" SpeechLLM on SQA task while other models are not instruction-tuned which gives some advantages to SpeLangy? To answer this question and the first point I made in the last section, I think the author needs to give a clear definition of *speech-text pretraining*. Starting from there, we might need to revisit the entire story.

Given these I am rating this paper 4 and would be happy to discuss more with the authors.

---

> ### Author Response · Authors · 2025-11-22
> **Rebuttal**
>
> We thank the reviewer for recognizing the comprehensiveness of our controlled ablations and the significance of our data-driven analysis that goes beyond standard benchmark comparisons. We address your concerns about task scope, model details, and fairness of comparisons below.
>
> > **W1. Is SQA representative? There is a need to show SQA performance correlates with speech-text pretraining quality.**
>
> This is a great question. We choose SQA due to three primary reasons:
> - It is a *core real-world use case* (voice assistants, spoken search, tutoring, etc.).
> - It provides a *mature, standardized evaluation protocol* with multiple benchmarks and strong baselines.
> - It allows a *clean experimental setup*: we use only interleaved speech–text continued pretraining, avoiding additional ASR/S2T/TTS losses that could confound the effect of data curation (task interference, data-mix ratios, etc.).
>
> To address the concern about generality, we show that indeed optimizing for SQA during pretraining also transfers to real-world chat evaluations like Alpaca-Eval. We conduct a very lightweight SFT stage from our pretrained checkpoints (details in the updated draft in section 3.7) on mixed speech–text conversational data and evaluate responses using LLM-as-judge metrics for *both text and audio quality* (using win-rates, details in appendix L of the updated draft). We use three pretrained checkpoints:
>
> - *Coarse*: coarse interleaved baseline only trained on web-crawled data (row 1 of table 1 in draft).
> - *Fine*: fine interleaved model only trained on web-crawed data (row 2 of table 1 in draft).
> - *Fine + Syn*: fine interleaved model trained with Quest data (best model in table 2 of the draft).
>
> We provide results in the table below (and have added them as table 5 in the revised draft).
>
> | Pretrain ckpt |      Text quality       |               |         Audio Quality          |        |        |        |        |
> |--------------:|------------------------:|--------------:|:------------------------------:|:------:|:------:|:------:|:------:|
> |               | spoken-alpaca          | noisy-alpaca  | Eval 1                         | Eval 2 | Eval 3 | Eval 4 | Eval 5 |
> | coarse        | 42.6                   | 45.2          | 37.4 (17.2)                    | 33.3 (24.1) | 34.3 (18.1) | 37.0 (16.3) | 38.8 (16.9) |
> | fine          | 44.3                   | 47.3          | 39.9 (18.5)                    | 33.8 (23.7) | 36.4 (11.6) | 38.0 (16.9) | **41.9 (20.7)** |
> | fine + syn    | **47.4**               | **48.8**      | **41.1 (17.1)**                | **36.6 (23.1)** | **40.1 (18.7)** | **39.4 (16.9)** | 39.3 (16.8) |
>
> We indeed find that the models trained with **our best data configurations (fine interleaving and synthetic data) are preferred in the majority of pairwise comparisons** and **never underperform the baseline configuration**, indicating that higher SQA accuracy correlates with better downstream conversational quality after SFT.
>
> We have added these new results in the revised draft showcasing that our data choices indeed **generalize to real-world speech tasks**.
>
> > **Q1. Backbone LM details**
>
> The backbone LM we start from follows the same pretraining recipe and uses the same pretraining data as the base model outlined in ([Li et al, 2025](https://arxiv.org/abs/2507.13575) Section 4.3). It has undergone no additional continued-pretraining or post-training (SFT or RLHF). The LM does not support speech tokens natively in the pre-training stage. We extend the vocabulary to include speech tokens. We initialize the new speech token embeddings randomly with Xavier normal initialization. We apoligize for not providing these details earlier and thank you for highlighting this point. We have now spelled this out explicitly in the revised Sec 3.2.
>
> > **Q2. Speech tokenizer details**
>
> We use a 1B-parameter Conformer encoder with 8x downsampling. We trained the tokenizer jointly with a combination of ASR loss and reconstruction loss, to jointly optimize phonetic and higher-level structure. Our tokenizer produces tokens at 80 ms per token (12.5Hz). It supports multilingual inputs. We have added these details in Sec 3.2 of the revised draft.
>
> > **Q3. Baseline LM performance in ablation tables**
>
> Thank you for pointing this out and we apologize for the oversight. We have added the base-LM numbers (as *Text-Init*) to all our ablations tables in the revised draft (Tables 1–3), reporting performance on text-only benchmarks before any speech–text pretraining. Our results show that our speech-language pretraining phase slightly degrades Core-EN performance while improving upon MMLU performance. These results are as expected based on our text data mixture and prior results from ([Gunter et al, 2024](https://arxiv.org/abs/2407.21075)) showcasing that continued pretraining can induce different performance benefits on reasoning-intensive tasks like MMLU vs other standard language modeling tasks like PIQA / Webqs.

---

> ### Author Response · Authors · 2025-11-22
> **Rebuttal contd.**
>
> > **Q4. Is Quest pretraining effectively task-specific finetuning? Fairness of comparison?**
>
> Thank you for raising this important question.
>
> First, we would like to clarify that even though Quest is in a Q/A format in text, due to our sentence-level chunking procedure during the TTS conversion, the interleaved speech-text Quest samples are **not** explicitly formatted as audio-questions + text-responses. Hence, in our final Quest dataset, many samples are split over multiple segments thereby breaking the local Q/A format during speech-text interleaving.
>
>
> Second, we respectfully point out that other base SpeechLMs also include QA/instruction-like data during their pretraining procedure. **GLM-4-Voice** uses FineWeb-Edu / Chinese-FineWeb-Edu, which are known to **upweight Q/A-formatted content** and exam-style QA pairs (see [Wettig et al, 2025](https://arxiv.org/abs/2502.10341) and [Godey et al, 2025](https://arxiv.org/abs/2510.25771) for evidence of this). **Qwen-Audio / Qwen2-Audio** perform **multi-task pretraining** with explicit `<|question-answer|>` tags; QA format is directly in the pretraining mixture. Thus, our use of Quest as a structured QA-like corpus is aligned with existing practice in speech-text pretraining. Additionally, the language models that **Kimi-Audio**, **Qwen-Audio** and **Qwen-2-Audio** start training from are the Qwen language models, which have instruction tuning data in their pretraining data mixture (see sec 2.1 of [Bai et al, 2023](https://arxiv.org/abs/2309.16609)). Hence, they have already seen similarly formatted data during their language pretraining, whereas we start from a pure base LM that has no instruction tuning data in the pretraining mixture.
>
> Third, to further address fairness of comparison with only base models, we also compare SpeLangy against **SFT-tuned variants** of the baselines (e.g., Kimi-Audio-Chat, Qwen-Audio-Instruct, Qwen2-Audio-Instruct). We find that **SpeLangy matches or is competitive** with these SFT models on SQA. This showcases that our data recipes help improve SQA performance beyond simply instruction tuning and QA-like formatting.
>
> | Type | Model         | #Ps  | SWQ  | STQ  | SLQ  | Average |
> |------|---------------|------|------|------|------|---------|
> | Base | Kimi-Audio    | 10.5B| 44.0 | 33.8 | 47.0 | 41.6    |
> | Base | Qwen-Audio    | 8.4B | 45.7 | 30.3 | 46.0 | 40.7    |
> | Base | Qwen-2-Audio  | 8.4B | 45.7 | 33.4 | 47.0 | 42.0    |
> | Base | SpeLangy      | 3.8B | 45.7 | 44.6 | 65.0 | 51.8    |
> | SFT  | Voxtral-Mini  | 4.7B | 41.6 | 46.6 | 65.3 | 51.2    |
> | SFT  | GLM-4-Voice   | 9.9B | 43.3 | 52.4 | 64.7 | 53.4    |
> | SFT  | Kimi-Audio    | 10.5B| 47.5 | 40.4 | 51.3 | 46.4    |
> | SFT  | Qwen-Audio    | 8.4B | 50.7 | 33.4 | 67.0 | 50.3    |
> | SFT  | Qwen-2-Audio  | 8.4B | 50.7 | 40.5 | 62.3 | 51.2    |
>
> However, we agree with the reviewer that the definition of speech-text pretraining might be imprecise due to the blurry boundary between pretraining and early instruction-like formatting in modern large-scale pipelines ([Su et al, 2025](https://arxiv.org/abs/2412.02595), [Ding et al, 2025](https://arxiv.org/abs/2504.18425), [Liu et al, 2025](https://arxiv.org/abs/2507.13264))
> To make our scope more precise, we have added more text in the last paragraph of the introduction (in blue) to clearly specify our work's motivation, goal and scoping. Indeed, we think it is an important research question to study more effective methods to bridge the gap between speech-text pretraining and task-specific fine-tuning, and hope future works address such questions.
>
> **We once again thank the reviewer for their efforts in carefully evaluating our paper, it is greatly appreciated! We believe your suggestions have greatly improved the quality of our paper, and hope that our responses have sufficiently addressed your concerns. We look forward to your response and further discussion!**

---

> ### Comment · Reviewer_Yb68 · 2025-11-28
> **Response**
>
> Thanks the author for supplying more results/baselines which makes the paper more solid.
> Regarding the my concerns about whether SQA task well correlates with speech-text pretraining, the author explains the motivation from the perspective of real-world application and validates the transferability through experiments. That partially addresses the concern. However, as the author says, the confusion of speech-text pretraining still exists. It would have had a bigger impact if the paper can find a clean way to pre-train and qualitatively evaluate/iterate a pre-trained speech LLM (e.g. perplexity / scaling law curve text LM).
> Nevertheless, I agree with the author that SQA is a very essential and common application for speech LLM. Therefore, if limiting the scope within SQA, this paper is a good empirical study that many real-world speech LLM applications can benefit from. Thus I am updating my rating to 8.
>
> [11/27 edit] can't seem to edit the review as for now, will update once the review revision is open.

---

### Official Review · Reviewer_zgtr · 2025-10-31

**Soundness:** 3
**Presentation:** 3
**Contribution:** 3
**Rating:** 8
**Confidence:** 4

**Summary:**

This paper studies the design decisions applicable to text-audio interleaved pretraining. First, they study the granularity of interleaving that should occur given aligned text and audio and find that fine grained interleaving is superior. Second, they study the usage of synthetically aligned text-audio using TTS and existing text corpora and find the addition of synthetic data offers advantages due to topical biases in existing natural audio data. Finally, they study the method for sampling chunks of data given a possible interleaved sequence at a given chunk granularity. Finally, they combine their best decisions to adapt a text-backbone model into a competitive pretrained text-audio LLM.

**Strengths:**

- The work does well designed low-level ablations that lead to clear suggestions about the design space of Speech-Language Pretraining. This type of non-glamorous but important study seems extremely likely to be valuable to other practitioners in the space and enables the authors to train a strong model themselves!
- The paper goes above and beyond most works of any form in terms of experimental rigor, including running contamination analysis.
- The synthetic data study in addition to the domain analysis of speech Pretraining data offers valuable insights for speech training even beyond the specific modeling paradigm studied in the rest of the work, which feels likely to remain a useful finding for some time.

**Weaknesses:**

- The work primarily focuses on evaluations in which the model must generate text, but does not evaluate how these decisions impact the models ability to generate speech in either S->S settings or in TTS usage.
- The works evaluations of the whole system does not compare to the simplest baseline of pipelining the text-init with a common ASR system such as Whisper.
- For a data centric work, the work doesn't actually provide much in the way of details of what the original source 10M hours of audio data is. This makes it inherently unreproducible.

**Questions:**

- What is the source of the original 10M hours of raw audio?
- How is inference operationalized given the interleaved pretraining? Is each chunk surrounded by a special token to indicate the modality which should be generated? Or is logit masking performed in order to generate only tokens of one modality at inference time?
- Does training on synthetic data collapse the ability of the model to generate voices beyond those generated in the synthetic data? Does it have any negative impacts on held out validation data from the speech domain?
- This model should in theory be able to generate speech right? If so, is there any reason why you didn't benchmark on tasks such as sBlimp?

---

> ### Author Response · Authors · 2025-11-25
> **Rebuttal**
>
> We thank the reviewer for the very positive assessment of our experimental rigour and for highlighting the value of our data-centric analysis. We address your concerns about evaluations scope, baselines, and reproducibility below.
>
> > **W1, Q3, Q4. Evaluations on S→S tasks? Effect of synthetic data?**
>
> Thank you for this set of insightful questions. We wish to re-emphasize that the primary goal of our work is to conduct a clean, controlled empirical study of data-centric choices for improving spoken question-answering in the speech-to-text (S→T) setting. We focus on S→T for three reasons: (i) S→T benchmarks have more mature and structured evaluation protocols, which makes them well-suited for targeted ablations, (ii) producing correct text outputs is a necessary first step before meaningfully assessing speech-to-speech (S→S) generation, and (iii) S→S evaluation is considerably more delicate, as it can entangle semantics (what is said) with acoustics (how it is said). In particular, using ASR to convert S→S outputs back to text introduces additional error from the ASR system, whereas directly relying on speech-token log-likelihoods risks over-emphasizing acoustic fidelity relative to semantic correctness.
>
> We however agree with the reviewer that S→S evaluation is also an important dimension to measure. As a first step, we hence evaluated our models on Sblimp and StoryCloze (both Spoken [SSC] and Topic [TSC] variants). The results for each individual setting studied in our paper are presented below.
>
> | Interleaving Granularity | Sblimp  | SSC S->S | TSC S->S |
> |------|---------------|------|------
> | Coarse | 54.1    | 51.3 |73.2 |
> | Fine |  54.5  | 53.3 | 73.5 |
>
> | Data Mix | Sblimp  | SSC S->S | TSC S->S |
> |------|---------------|------|------
> | Web-crawl 100% | 54.5 | 53.3 | 73.5 |
> | Web-crawl 53% + Krist 47% | 54.6 |  52.7 | 74.1 |
> | Web-crawl 66% + Quest 34% | 54.4 |  51.6 | 73.0 |
> | Web-crawl 59% + Quest 6% + Krist 35% | 54.5 | 52.0 | 73.3 |
> | Web-crawl 40% + Quest 27% + Krist 33% | 54.4 | 51.8 | 73.2 |
>
> | Sampling scheme | Sblimp  | SSC S->S | TSC S->S |
> |------|---------------|------|------
> | Stochastic | 54.4    | 51.8 | 73.2 |
> | Deterministic |  54.5  | 51.7 | 69.1|
>
> We mainly find:
> 1. Fine interleaving outperforms coarse interleaving on both Sblimp and Storycloze evaluations. This emphasizes that our main takeaways regarding the interleaving strategy holds true even for speech-to-speech evaluations.
> 2. For the synthetic data variants, the results are not fully conclusive. We note that the Web-crawl 53% + Krist 47% checkpoint improves on TSC while degrading on SSC. For the other checkpoints, on SSC most of them seem to drop, whereas for TSC the drop seems to be much more minor.
> 3. For the deterministic vs stochastic experiment, we note that both checkpoints used synthetic data in the training mix, and due to this it is hard to draw conclusions as to whether there is a large improvement in favour of either strategy.
>
> Finally, we note that our synthetic data uses different voices from the natural evaluation data, which can affect audio-token log-likelihoods and thus these S→S scores. This is precisely the challenge we mentioned above regarding the coupling of semantics and acoustics in S→S evaluation. Nonetheless, our preliminary results suggest that synthetic data does *not* collapse the model’s speech generation capabilities, at most, it leads to mild degradations.

---

> ### Author Response · Authors · 2025-11-25
> **Rebuttal contd**
>
> > **W1, Q3, Q4 contd.  Evaluations on S→S tasks? Effect of synthetic data?**
>
> To further test this on a broader set of evaluations, we also conduct a very lightweight SFT stage from our pretrained checkpoints (details in the updated draft in section 3.7) on mixed speech–text conversational data and evaluate responses using LLM-as-judge metrics for *both text and audio response quality* (using win-rates, details in appendix L of the updated draft). We use three pretrained checkpoints:
>
> - *Coarse*: coarse interleaved baseline only trained on web-crawled data (row 1 of table 1 in draft).
> - *Fine*: fine interleaved model only trained on web-crawed data (row 2 of table 1 in draft).
> - *Fine + Syn*: fine interleaved model trained with Quest data (best model in table 2 of the draft).
>
> We provide results in the table below (and have added them as table 5 in the revised draft).
>
> | Pretrain ckpt |      Text quality       |               |         Audio Quality          |        |        |        |        |
> |--------------:|------------------------:|--------------:|:------------------------------:|:------:|:------:|:------:|:------:|
> |               | spoken-alpaca          | noisy-alpaca  | Eval 1                         | Eval 2 | Eval 3 | Eval 4 | Eval 5 |
> | coarse        | 42.6                   | 45.2          | 37.4 (17.2)                    | 33.3 (24.1) | 34.3 (18.1) | 37.0 (16.3) | 38.8 (16.9) |
> | fine          | 44.3                   | 47.3          | 39.9 (18.5)                    | 33.8 (23.7) | 36.4 (11.6) | 38.0 (16.9) | **41.9 (20.7)** |
> | fine + syn    | **47.4**               | **48.8**      | **41.1 (17.1)**                | **36.6 (23.1)** | **40.1 (18.7)** | **39.4 (16.9)** | 39.3 (16.8) |
>
> We indeed find that the models trained with **our best data configurations (fine interleaving and synthetic data) are preferred in the majority of pairwise comparisons** and **never underperform the baseline configuration**, indicating that higher SQA accuracy correlates with better downstream conversational quality after SFT. This further shows that pretraining with our synthetic datasets also induces better audio generation capabilities after a short post-training phase.
>
> We have added these new results in the revised draft showcasing that our data choices indeed **generalize to real-world speech tasks**. However, as noted earlier, it is indeed an important avenue for future work to investigate the impact of data curation recipes for speech to speech tasks during pretraining.
>
> We thank the reviewer again for raising this important point and have expanded on this in our limitations and future works section, by adding a section on "End-to-end evaluation".
>
> > **W3, Q1. More details and source of web-scale training data.**
>
> We thank the reviewer for raising this important point, and apologize for our oversight in providing more details earlier.  All our web-crawled speech data comes from publicly available podcast RSS feeds and similar spoken-word streams. We do not scrape behind paywalls and avoid clearly copyrighted catalogue content such as commercial audiobooks and music albums. Our crawler respects `robots.txt` directives and website-specific terms-of-use. Our data was specifically curated to be diverse, conversational, multilingual, and multi-speaker. Audio samples average approximately 30 minutes in duration, with around 60% consisting of English speech. We have added these details in Appendix B of the revised paper.
>
> > **Q2. More details on inference mechanism.**
>
>  During pretraining we wrap audio spans in dedicated boundary tokens (e.g., `<start_of_audio>`, `<end_of_audio>`), but **do not apply additional logit masking** between modalities. For **SQA evaluation**, we compute **log-likelihood only over the reference text tokens**, ignoring the audio token log-probs. For generative use, the model learns when to switch modalities from the training statistics---providing `<end_of_audio>` in the context reliably steers it to text generation without explicit masking.
>
> > **Cascaded baseline of Whisper ASR + text-init LM.**
>
> We thank the reviewer for suggesting this baseline. We agree this would better contextualize our numbers to a strong upper-bound system. We are currently running the evaluations and will add the numbers to the revised paper and update them here as soon as possible.
>
> **We once again thank the reviewer for their efforts in carefully evaluating our paper, it is greatly appreciated! We believe your suggestions have greatly improved the quality of our paper, and hope that our responses have sufficiently addressed your concerns. We look forward to your response and further discussion!**

---

> ### Comment · Reviewer_zgtr · 2025-11-25
>
> The added experiments are appreciated! I want to re-iterate that I find this work to contain useful findings with clear experiments and to be overall strong, seemingly in contrast to the other 3 reviewers.
>
> The additional experiments reinforce my belief that this work is useful, however, given the discrete choices available, I will maintain my score, which was already on the clear positive side. Given the still somewhat opaque details about the source of some the data underlying this model training process and lack of releases of usable resources (code/models/data/) this work is meaningfully not within the 10 category in my assessment since it is inherently non-reproducible.

---

> > ### Author Response · Authors · 2025-11-27
> > **Response**
> >
> > We thank the reviewer for their strong support of our paper, recognizing the strength of our empirical contributions and main takeaways. We have updated the paper with the new results of the cascaded baseline comparison as well. We truly appreciate your contributions in helping us improve the overall strength of our draft.

---

### Author Response · Authors · 2025-11-23
**General Response to All Reviewers**

**We thank all reviewers for their thoughtful and constructive feedback, and for highlighting several key strengths of our work: the systematic, data-centric ablation design and clean experimental setup (zgtr, Yb68, jiGx), the breadth and rigor of our empirical evaluation (zgtr, QDUW, jiGx), the clarity and organization of the paper (Yb68, QDUW, jiGx), and the importance of studying data curation as an under-explored but impactful direction (QDUW, jiGx)**.

Following the reviews, we have made the following changes to the draft. All changes are highlighted in blue.
- We have added additional clarifying text to emphasize the scope of our empirical setup and evaluations in the last paragraph of the introduction (section 1).
- We have provided additional details on the base-LM and speech tokenizer (section 3.2).
- We have added the base-LM text-only results to all the ablation tables (Table 1, 2 and 3) to better contextualize our ablation results.
- We have updated the text in section 3.6 to clarify the audio-loss-masked setting.
- We have added a section 3.7 on "Our data-centric lessons transfer after post-training": this section provides additional results showcasing that our data methods also improve on more instruction following text and audio quality metrics. In Appendix M, we provide further details about the post-training setup.
- We have moved the contamination proportion table from the appendix to table 7.
- We have added an ethics statement in Appendix B, clarifying our data sourcing strategies.
- We  have updated the limitations and future work discussion in Appendix N to highlight further evaluations (including few-shot and speech-to-speech settings) that we view as important directions for future work.

---

### Author Response · Authors · 2025-12-04
**Final Comments by Authors (Summary for AC)**

Dear Area Chair,

Thank you for your time in handling our submission. To help your decision, we briefly summarize the main strengths highlighted by the reviewers and the key revisions/new evidence added during rebuttal.

---

## 1. Strengths Highlighted by Reviewers

- **Systematic data-centric study** of speech–text pretraining with clean, controlled ablations on interleaving, synthetic data, and sampling (zgtr, Yb68, QDUW, jiGx).
- **Strong empirical rigor**: careful contamination checks, KL-divergence and topic analyses, and comparisons to strong SpeechLM baselines (zgtr, QDUW, jiGx).
- **Clarity and organization** of the paper and schematics (Yb68, QDUW, jiGx).
- **Practical impact**: lessons that directly inform how to build SpeechLMs and a 3.8B model (SpeLangy) that outperforms much larger SpeechLMs on SQA (all reviewers).

---

## 2. Key Revisions and New Results

1. **Scope and SQA representativeness (Yb68, QDUW, jiGx)**
   - Clarified that our goal is a *clean SQA-focused speech–text pretraining study*.
   - Added **post-training chat evaluations** (LLM-as-judge on text+audio) showing that configurations that perform best on SQA are also preferred in real-world conversational quality.

2. **Speech-to-speech evaluation (zgtr, jiGx)**
   - Added **preliminary S→S results** on sBlimp and Spoken/Topic StoryCloze.
   - Fine interleaving consistently outperforms coarse, confirming that our main interleaving lesson holds for S→S as well.

3. **Baseline coverage and cascades (zgtr, Yb68)**
   - Added a **Whisper-v3-large + text-LM cascade baseline**.
   - As expected the cascade remains strongest, but the 3.8B SpeLangy **substantially narrows the gap**, especially on SWQ, illustrating progress toward competitive end-to-end SpeechLMs.

4. **Definition and fairness of "speech–text pretraining" (Yb68, QDUW)**
   - Clarified that **Quest** data, after sentence-level chunking, is *not* fed as explicit "audio-question/text-answer" pairs and is used only via a generic LM objective.
   - Noted that other "base" SpeechLMs (GLM-4-Voice, Qwen-Audio, Qwen2-Audio) also include QA/instruction data at pretraining time.
   - Added comparisons to both **base and SFT-tuned** versions of these models; SpeLangy remains competitive or stronger on SQA despite being much smaller.

5. **Model details, base-LM performance, and ethics (Yb68, QDUW, zgtr)**
   - Added details on the **base LM** and **Conformer speech tokenizer**, and included **base-LM text-only rows** in all ablation tables to show the impact of speech–text pretraining.
   - Surfaced the contamination table into the main text and added an **ethics statement** describing data sourcing and compliance.

---

## 3. Post-Rebuttal Reviewer Stance

- **zgtr**: Maintains a **score of 8**, emphasizing clear experiments and strong rigor, and views the added experiments as reinforcing the usefulness of the work.
- **Yb68**: Raises the score to **8**, stating that, within the SQA scope, this is a **valuable empirical study that real-world speech LLM applications can benefit from**.
- **QDUW**: Describes the paper as **comprehensive and well-executed**; while cautious about scope, explicitly notes they **"would not mind if the paper is accepted."**
- **jiGx**: Continues to highlight that our work **fills a gap in controlled data ablations for SpeechLMs** and demonstrates **practical value** through SpeLangy.

---

## 4. Why We Believe the Paper Meets the ICLR Bar

- It provides **rare, carefully controlled data-centric ablations** for SpeechLMs, an area where large-scale pretraining practices are typically opaque.
- It offers **concrete, adoptable recipes** (fine interleaving, high-quality synthetic QA data, deterministic sampling) backed by analysis, not just single-point benchmarks.
- It delivers **meaningful performance gains**: a 3.8B SpeechLM that **outperforms much larger SpeechLMs** on SQA, substantially closes the gap to a strong cascade, and whose data lessons **transfer after post-training**.
- All major reviewer concerns (scope, S→S, baselines, fairness, and details) have been **directly addressed** with new experiments and clarifications; no unresolved technical flaws remain, and two reviewers now give **clear accept (8)** scores.

We appreciate your careful consideration and hope this concise summary is helpful in your decision.

Best regards,
*Authors of Submission 15267*

---

### Meta-Review · Program_Chairs · 2026-01-06

**Summary:**

This paper presents a data-centric investigation into speech–language pretraining, targeting the development of SpeechLLM. Reviewers initially raised concerns regarding the narrow evaluation focus on SQA, the incremental nature of some findings, the fairness of comparisons to prior models, and the insufficient analysis of modest gains. These concerns have been largely addressed in the rebuttal through the addition of new downstream conversational and evaluation demonstrating transfer beyond SQA, positioning its contribution as controlled ablations under web-scale conditions, and expanded comparisons against both text-only and SFT/S2S-capable speech models. The authors also strengthen the empirical narrative by providing mechanistic analyses (e.g., KL-divergence between speech- and text-conditioned outputs, chunk length statistics) and enhancing clarity and presentation. While the work remains primarily empirical, it offers well-justified and practically valuable insights. Since the overall responses sufficiently resolve the reviewers’ concerns, the AC places the paper above the acceptance threshold.

**This paper is conditionally accepted provided the authors do the following for the camera-ready**:
[Ethics] The authors must include discussion of data provenance, consent, and compliance in the ethics statement and appendix as promised in the rebuttal.

**Reviewer Concerns:**

1. Limited and potentially unrepresentative evaluation scope (R1-R4)
2. Insufficient methodological and data transparency affecting reproducibility (R1, R2 and R4)
3. Fairness and appropriateness of baselines and comparisons (R1, R2, and R3)
4. Limited novelty or depth of insight from some findings (R3 & R4)

See further elaboration below (Reviewer Scores).

**Reviewer Scores:**

Both R1 and R2 find the rebuttal responses satisfactory (prior to the termination of the discussion stage), and maintain a very positive rating of 8 for this work.

The AC feels that R3 would likely find the rebuttal satisfactory enough to raise the rating (probably 4 -> 6), primarily because R3's main challenge is the contribution and evaluation breadth, not fatal technical flaws. And, the authors directly addressed every substantive critique with new experiments, clearer positioning, and better framing, rather than deflection or hand-waving. Similarly, the responses to R4 provide new evaluations, improve interpretability, and fix presentation issues. The AC also believes R4 would at least raise the rating to 6.

---

### Decision · Program_Chairs · 2026-01-26

**Decision:**

Accept (Poster)

**Comment:**

Conditions for acceptance have been satisfied.